# Context-specific effects of sequence elements on subcellular localization of linear and circular RNAs

Maya Ron[1] & Igor Ulitsky [1✉]

Long RNAs vary extensively in their post-transcriptional fates, and this variation is attributed in part to short sequence elements. We used massively parallel RNA assays to study how sequences derived from noncoding RNAs influence the subcellular localization and stability of circular and linear RNAs, including spliced and unspliced forms. We find that the effects of sequence elements strongly depend on the host RNA context, with limited overlap between sequences that drive nuclear enrichment of linear and circular RNAs. Binding of specific RNA binding proteins underpins some of these differences—SRSF1 binding leads to nuclear enrichment of circular RNAs; SAFB binding is associated with nuclear enrichment of pre-dominantly unspliced linear RNAs; and IGF2BP1 promotes export of linear spliced RNA molecules. The post-transcriptional fate of long RNAs is thus dictated by combinatorial contributions of specific sequence elements, of splicing, and of the presence of the terminal features unique to linear RNAs.

[1] Departments of Biological Regulation and Molecular Neuroscience, Weizmann Institute of Science, Rehovot 76100, Israel. ✉email: igor.ulitsky@weizmann.ac.il

Eukaryotic cells are divided into two main compartments: the nucleus where RNAs are made and processed, and the cytoplasm where some RNAs are translated into proteins. Only a subset of the RNA molecules that are made are efficiently exported from the nucleus to the cytoplasm (e.g. mRNAs, tRNAs, and rRNAs), other RNA molecules, including many lncRNAs, are enriched in the nucleus[1,2]. Nuclear export of long RNAs is generally thought to be dependent on factors that recognize a 5′ cap[3]. Surprisingly, circRNAs also appear to mostly localize to the cytoplasm[4,5], but there are also examples of circRNAs which function in the nucleus[6,7]. It is not known if the cytoplasmic presence of circRNAs results from an active export process or from their high stability[8,9] that allows their escape into the cytoplasm during mitosis in the course of cell division. The function of the vast majority of circRNAs remains unknown[10], and it is unclear how many circRNAs are functional[11,12], but it is likely that those that do carry out influential functions require specific subcellular distribution patterns. Furthermore, it is possible that aberrant circRNA accumulation, which may occur during aging or neurodegeneration[13,14], can have localization-dependent detrimental effects.

Beyond the nuclear and the cytoplasmic compartments, very little is known about localization of circRNAs to other compartments. In a recent study which systematically studied circRNA representation in subcellular fractions of HepG2 cells[15], it was shown that circRNAs are widely distributed among different subcellular fractions (nucleus, cytoplasm, mitochondria, ribosome, cytosol, and exosome), and that circRNAs in the different compartments had different characteristics in terms of length and G/C content. In addition, perturbations of different export factors in Drosophila DL1 and human HeLa cells have shown that the lengths of mature circRNAs dictate the mode of their nuclear export[16]. In neurons, some circRNAs were shown to localize to synapses[17], but the elements that dictate such localization are unknown.

During splicing, introns are removed from the pre-mRNAs by the spliceosome to generate the mature mRNA. Splicing and polyadenylation of the pre-mRNA are typically rapid and mostly cotranscriptional, and are carried out by proteins which sometimes remain bound to the processed RNA. By comparing the localization and quantity of spliced and unspliced RNAs in the different cell compartments it was shown that splicing enhances the export of RNAs from the nucleus to the cytoplasm[18–20], and that splicing factors can facilitate nuclear export by recruitment of export factors to the mature mRNA[18,20,21]. It is unknown if the splicing involved in formation of circRNAs contributes to their export.

The localization of long RNAs in the cells is regulated by *trans* and *cis*-regulatory factors. Massively parallel RNA assays (MPRNAs) were recently introduced as a tool to study the contribution of short *cis*-acting sequences to retention of lncRNA in the nucleus or on chromatin, or to efficient export of single-exon RNAs[22–25]. All these studies were based on an unspliced 'host' mRNA carrying the variable sequences in its 3′UTR. Cells expressing the inserts were fractionated into chromatin, nuclear or cytoplasmic fractions, and the fragments from each compartment were sequenced. Oligos sequences that are enriched in one of the compartments were considered to drive export or nuclear retention. There was some overlap between the identified motifs[22,23], but many motifs were observed in one study but not in others, possibly due to the differences in their experimental design. A key question that we set out to systematically address here is what is the contribution of the host RNA sequence and form to the effect that short sequence elements have on RNA localization and its stability.

## Results

### Characterization of circRNA localization in MCF-7 cells. In order to map the landscape of subcellular localization of

circRNAs in MCF-7 breast cancer cells, we used the experimental RPAD sequencing protocol that enriches for circRNAs[26] on nuclear and cytoplasmic fractions (Supplementary Fig. 1a). We used these data to quantify the localization of circRNAs defined by 94,752 previously annotated back-splicing junctions, out of which 3458 junctions were supported by at least 10 read pairs in at least one sample, and corresponded to circRNAs with no more than five exons and no longer than 5 kb in length. As expected from previous studies, circRNAs were largely enriched in the cytoplasm, though to a substantially varying degree (Supplementary Fig. 1b). We validated the localization of selected circular and nuclear circRNAs (Supplementary Fig. 1c). Combining the RPAD-based quantification with manual inspection and analysis of data from HepG2 cells[15], we selected 28 relatively cytoplasmic circRNAs, 17 relatively nuclear ones, and 37 circRNAs with some previous functional characterization (Supplementary Data 1). We designed a library composed of 140 nucleotides (nt) fragments that tile the exonic sequences of these circRNAs ('CircLibA') (Supplementary Fig. 1b and Supplementary Data 1). We also used 'NucLibA'—a previously developed library of 110 nt tiles from the exonic sequences of human nuclear lncRNAs[22]. Most experiments were performed using both libraries side by side.

### Characterizing the influence of RNA contexts on the contribution of short sequences to RNA localization. To identify sequences that can affect subcellular localization when expressed in different RNA contexts, we used a massively parallel RNA assay (MPRNA); we cloned CircLibA and NucLibA libraries into expression vectors that produce three different RNA forms (Fig. 1a): (1) Linear spliced RNA—produced from the WT β-globin expression vector, a commonly used model for RNA localization[27], which has two introns that are spliced in the nucleus. (2) Unspliced RNA—produced from the β-globin-Δintrons expression vector[27], which contains the human β-globin gene where both introns were removed, and therefore it is not spliced. The RNA produced from this vector was previously shown to be much less efficiently exported than WT spliced β-globin mRNA[18,28] (Fig. 1b). NucLibA and CircLibA libraries were cloned into the 3′UTR of the WT and Δintrons β-globin gene. (3) Two Circular RNA forms—based on an expression vector specifically tailored for expression of circular RNAs, where the sequences flanking the circularized exon are derived from intronic regions from the human ZKSCAN1 gene, which is known to form a circRNA by base-pairing of the intronic repeats and backsplicing[29]. We cloned the library into the middle of the circPVT1[30] sequence or a scrambled version of circPVT1 (SCRcircPVT1) (Fig. 1b). These backbones were used because shorter exonic sequences did not circularize efficiently.

Plasmid pools were transfected into MCF-7 cells that were fractionated after 24 h, and the Nuclear/Cytoplasmic (Nuc/Cyto) ratios of the library were quantified by qRT-PCR alongside the GAPDH and MALAT1 transcripts to evaluate fractionation efficiency (Supplementary Fig. 1d). In addition, we validated that the reporter mRNAs were efficiently spliced (Supplementary Fig. 1e). We then sequenced the inserts and determined the effect on localization by calculating the Nuc/Cyto ratio for each fragment (Fig. 1c, d). Importantly, when amplifying the circRNA libraries we used a reverse primer that spans the junction of circPVT1, that would not amplify the linear form, and so sequences that affect circularization frequency should not affect our results (Supplementary Fig. 1e). In addition, for each experiment, the expression levels are normalized to the overall expression of all the tiles in the same fraction, and so the Nuc/Cyto ratios we used always refer to the *relative* changes when compared with the baseline localization of each backbone

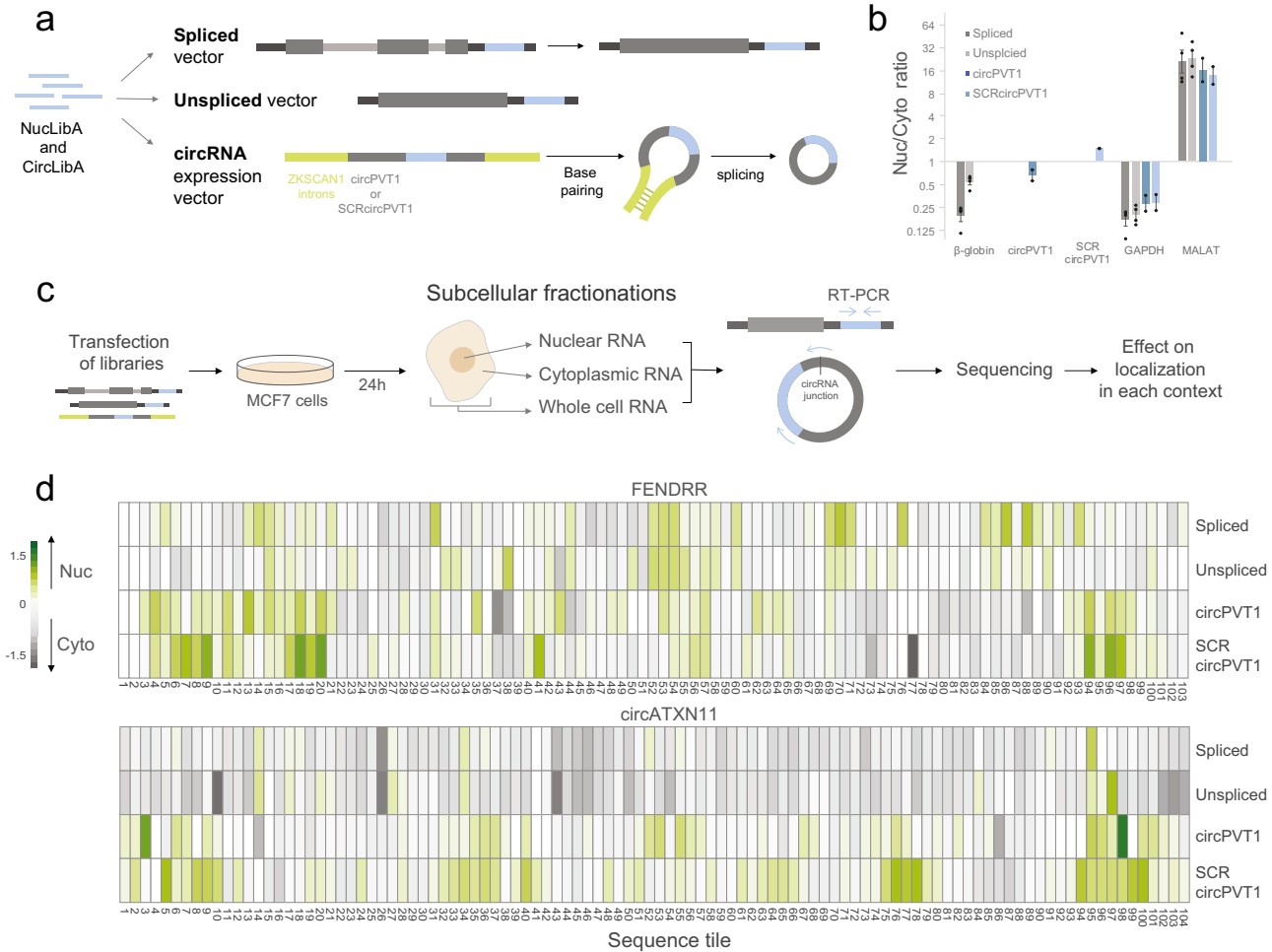

**Fig. 1 MPRNA for localization using different forms of RNA. a** CircLibA and NucLibA were cloned into four expression vectors; in the 3′ UTR of WT β-globin (spliced) and β-globin-Δintrons (unspliced), and in the middle of circPVT1 or SCRcircPVT1 (circular). **b** qPCR analysis of nuclear and cytoplasmic fractions following transfection of β-globin plasmids or circular expression vectors. $n = 4$ biologically independent samples, $n = 2$ for circular expression vectors. Data are presented as mean values $+/-$ SEM. **c** Outline of the experimental procedure; libraries were transfected, and cells were fractionated into nuclear and cytoplasmic fractions after 24 h. Libraries for sequencing were generated from total, nuclear and cytoplasmic RNA. **d** Subcellular localization of the tiles derived from the lncRNA *FENDRR* and the circRNA *circATXN11* in the spliced, unspliced, circPVT1, and SCRcircPVT1 contexts. Color indicates the $\log_2(\text{Nuc/Cyto})$ of each tile. Cytoplasmic shift is in gray, nuclear shift in green. Source data are provided as a Source data file.

(Fig. 1b). In the same experiment, we sequenced inserts from whole-cell extract (WCE), which allowed us to measure the expression levels conferred by each fragment.

**Context-specific effect of sequences of host RNA localization.** When considering all 5471 tiles that could be quantified from the combination of CircLibA and NucLibA, the effects of sequences on the subcellular localization of the host RNA varied substantially between contexts (Fig. 2a and Supplementary Fig. 2a). When examining the overall correlation between the localization of tiles in each context, we observed a higher correlation between the two linear forms (spliced and unspliced, Spearman's $R = 0.59$) and between the two circular forms (circPVT1 and SCRcircPVT1, $R = 0.42$), than between the circular and the linear forms (Fig. 2b, $R = -0.02$ to $R = -0.16$). In order to identify common features in tiles that are localized to the different compartments, we compared tiles that were significantly enriched in the cytoplasmic ($\log_2(\text{Nuc/Cyto}) < -0.3$, roughly corresponding to a 25% difference, and $P < 0.05$) or in the nuclear ($\log_2(\text{Nuc/Cyto}) > 0.3$ and $P < 0.05$) fractions of each library (Supplementary Fig. 2b). The G/C content of cytoplasmic tiles in the linear libraries (spliced and unspliced) was significantly higher than the G/C content of

nuclear tiles and all other tiles (Cyto vs. all other tiles: $P = 10^{-35}$ for spliced and $P = 5.2 \times 10^{-39}$ for unspliced), while in the circular form there is almost no difference between these groups (Fig. 2c). In addition, tiles driving cytoplasmic enrichment in the unspliced, and to a lesser extent in the linear spliced forms were predicted to be more structured ($P = 7.9 \times 10^{-38}$ for cytoplasmic tiles vs. all other tiles in the unspliced context, $P = 2.4 \times 10^{-10}$ for cytoplasmic tiles vs. all other tiles in the spliced context, Supplementary Fig. 2c), though it is difficult to uncouple the differential G/C content from the potential to form more stable structures, as shuffled sequences of the tiles driving cytoplasmic enrichment were also predicted to be more structured than shuffled sequences of other tiles (Supplementary Fig. 2c). The contribution of G/C context to cytoplasmic export in linear RNAs is consistent with our previous results that used another library, 'CytoLib', in the same introness β-globin context[25]. Differences in G/C content between NucLibA and CircLibA (44% for NucLib on average vs. 48% for CircLibA on average) also likely underlie the slightly different distributions of Nuc/Cyto ratios for tiles derived from the two libraries (Supplementary Fig. 2d).

Since the β-globin and the circular backbones had different sequences, we also conducted MPRNAs using sequence-matched

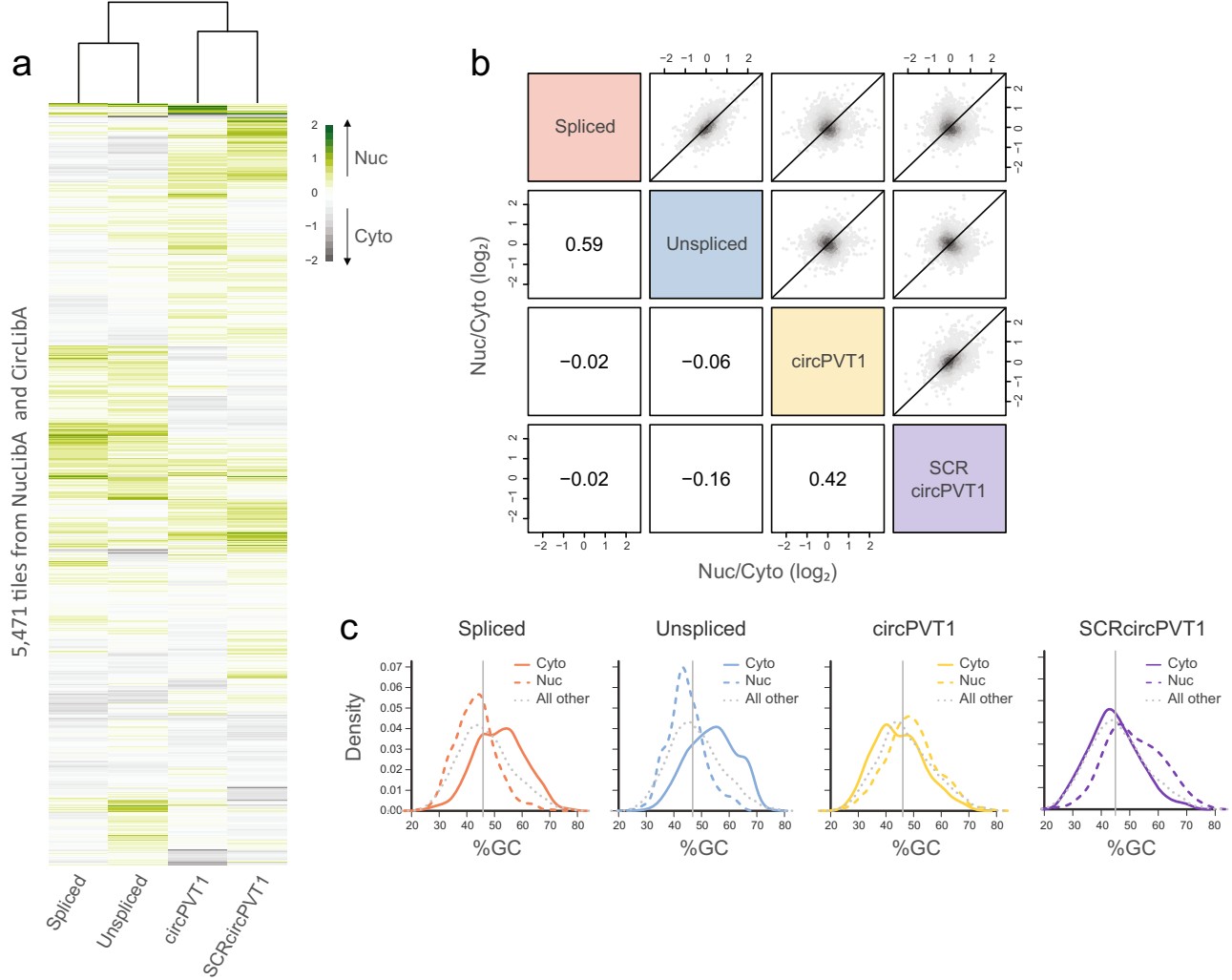

**Fig. 2 Parameters of tiles that drive localization to the different cell compartments. a** Subcellular localization of all tiles in the spliced, unspliced, circPVT1, and SCRcircPVT1 contexts. Color indicates the median $\log_2(\text{Nuc/Cyto})$ of each tile. Cytoplasmic shift is in gray, nuclear shift is in green. **b** Correlations between the localization of tiles in spliced, unspliced, circPVT1, and SCRcircPVT1 contexts. **c** G/C content distributions of tiles enriched in the cytoplasmic fraction (solid line), nuclear fraction (dashed line), or all other tiles (dotted line) in each context. Vertical gray line indicates the median of all tiles in the sample. Source data are provided as a Source data file.

backbones. We examined the possibility of circularizing the β-globin sequence, but found that it would not circularize efficiently in the ZKSCAN1 vector (Supplementary Fig. 3a). Circularization could be improved only by adding at least 30 flanking bases from circPVT1 (Supplementary Fig. 3b), and so we did not further pursue this strategy. Instead, we generated a backbone with a linear version of circPVT1 and used it for an MPRNA using the NucLibA and CircLibA libraries. Nuc/Cyto ratios in the "linear circPVT1" backbone were similar to those in the linear β-globin backbones and were dissimilar to the circular ones (Supplementary Fig. 3c), and so we conclude that the form of the backbone (linear/circular) is more influential than the sequence of the backbone when considering the effects of short sequence elements on subcellular RNA distribution.

When considering tiles originating from individual genes as a group, several groups exhibited specific effects (Supplementary Fig. 4a). Specifically, tiles derived from the human MALAT1 lncRNA were effective in driving nuclear enrichment in the linear spliced, and to a lesser extent in the unspliced context (Supplementary Fig. 4b). NucLibA also contained tiles derived from the sequences of MALAT1 orthologs in four other species. Mouse Malat1-derived sequences were also effective, but to a

lesser extent, whereas orthologous sequences from the more distal species—lizard, chicken, and zebrafish, appeared to be much less effective (Supplementary Fig. 4b). Specific regions within MALAT1 were previously shown to drive nuclear enrichment[23,31]. Interestingly, short tiles overlapping these MALAT1 regions were generally as effective as other MALAT1-derived tiles, suggesting that the information driving the strong nuclear enrichment of this lncRNA, which is found almost exclusively within nuclear speckles, is broadly distributed throughout the RNA, and has perhaps adapted specifically to be effective within human cells (Supplementary Fig. 4c, d), which may explain its poor sequence conservation between distal species[32].

**Context-specific effects of sequence elements on RNA stability.** Short sequence elements can also affect host RNA stability[33,34] which can influence Nuc/Cyto ratios. In order to test the effects of sequences in different contexts on RNA stability, we used actinomycin D (ActD) to inhibit transcription following plasmid transfection, extracted RNA at different time points, and sequenced the tiles. Then, we used the expression levels to calculate the half-life time of each fragment at each context. Since we

observed high correlation between the Nuc/Cyto ratios of tiles in the two circRNAs contexts (circPVT1 and SCRcircPVT1), the following experiments were conducted using only circPVT1 expression vectors.

The short sequence tiles had a substantial effect on the stability of the transcript. Tiles expressed as circular transcripts tended to be more stable than the linear form, similar to endogenous circRNAs[35,36], and among the linear transcripts, those unspliced tended to be relatively unstable (Supplementary Fig. 5a). Tiles affected stability differently when placed in the linear and circular context (Supplementary Fig. 5b), similar to what we observed when examining localization. Furthermore, there was overall limited correlation between the effects of tiles conferred on stability and localization (Supplementary Fig. 5c), explaining no more than 15% of the variance between tiles in their effects on localization. Higher stability was associated with a more cytoplasmic localization only for circular RNAs, supporting the notion that the circRNAs that are more cytoplasmic are ones that are more stable.

We also examined the correlation between G/C content of the tile and host RNA stability. Consistent with a previous study, using a different reporter system[34], we found that higher G/C content was associated with reduced RNA stability, mostly in the spliced linear context, and to a lesser extent in the circular context (Supplementary Fig. 5d).

**Signatures of RNA binding protein binding enriched in elements driving subcellular localization.** In order to identify RBPs that have a role in RNA localization, we examined eCLIP data from the ENCODE project[37,38] alongside analysis of known RBP binding preferences using AME, STREME[39], and RNAComplete profiles[40]. Based on this analysis, we selected four RBPs that appeared to be enriched in tiles exhibiting different Nuc/Cyto ratios, and so we hypothesized that they might regulate RNA localization—SRSF1, SAFB, IGF2BP1, and IGF2BP2 (Supplementary Fig. 6a).

In order to investigate the role of these RBPs in the regulation of RNA localization, we knocked down (KD) each of them in MCF-7 cells (Fig. 3a and Supplementary Fig. 6b) and performed MPRNAs with the linear spliced, unspliced, and circular expression vectors. We then fractionated the cells and sequenced the inserts in the KD and control cells (Supplementary Fig. 6c).

**SRSF1 binding drives nuclear enrichment of circular RNAs.** SRSF1 is an SR protein that is predominantly located in the nucleus. It binds purine-rich RNA sequences[38,41], has roles in RNA splicing[42], and was reported to have a function as an export adapter that is involved in mRNA nuclear export through the NXF1/TAP pathway[43]. Tiles bearing the SRSF1 binding motif were enriched in the nuclear fraction especially when expressed in either of the circular backbone sequences (Fig. 3b and Supplementary Fig. 6a), and they became more cytoplasmic upon SRSF1 KD in the circular form, and to a lesser extent in the linear spliced form, although these sequences were generally less nuclear in this context (Fig. 3c, d). We did not observe any effect of SRSF1 KD on the localization in the unspliced context. This suggests that SRSF1 is primarily involved in the regulation on the localization of spliced RNAs, and the binding of SRSF1 retains circRNA in the nucleus.

In order to evaluate the effects of SRSF1 on nuclear enrichment of circRNAs containing SRSF1 eCLIP clusters and/or predicted binding motifs, we selected 10 circRNAs based on the average number of SRSF1 binding motifs and average number of CLIP clusters across tiles, and evaluated their Nuc/Cyto ratios in SRSF1

KD and control cells. SRSF1 KD led to a significant cytoplasmic shift in Nuc/Cyto ratios for six of the circRNAs (Fig. 3e).

In addition, it was shown that SRSF1 (SF2) restricts circRNA expression in Drosophila[29], and KD of SRSF1 caused elevation in the expression levels of Laccase2 circRNA. We hence examined the association between change in localization and change in expression upon KD of SRSF1, and observed increased expression levels mainly in the circular context, with stronger effect on the tiles containing SRSF1-bound regions in eCLIP data (Supplementary Fig. 7a). We also observed an elevation in the expression levels of selected SRSF1-bound endogenous circRNAs upon SRSF1 knockdown (Supplementary Fig. 7b).

In order to evaluate the transcriptome-wide effect of SRSF1 on circRNA localization, we used RPAD to characterize circRNA expression in the nucleus and cytoplasm of MCF-7 cells following transfection of siRNAs targeting SRSF1 or a control. As expected from our MPRNA results, presence of SRSF1 binding motifs in the circRNA sequence was associated with increased nuclear localization in control cells (Supplementary Fig. 7c), and KD of SRSF1 led to a reduction in Nuc/Cyto ratios of bound circRNAs (Fig. 3f). Sequences of circRNAs with a >2-fold cytoplasmic shift in their localization upon SRSF1 KD were significantly enriched with SRSF1 binding motifs compared to those with a 2-fold nuclear shift (Supplementary Fig. 7d, e).

**SAFB binding drives nuclear enrichment of unspliced RNAs.** SAFB is an RNA binding protein that localizes to the nucleus, and is considered to be part of the 'nuclear matrix'[44]. It also binds purine-rich motifs, and it was shown to have a function in RNA processing and interaction with chromatin-modifying complexes[45,46]. In MCF-7 cells, an iCLIP experiment showed strong enrichment of SAFB binding in non-coding RNAs[45], including MALAT1, an unspliced lncRNA that is heavily represented in NucLibA.

We observed enrichment of tiles containing the SAFB binding motif in the nucleus when expressed in the linear spliced or unspliced form (Fig. 4a and Supplementary Fig. 6a), but these tiles became less nuclear after SAFB KD only in the unspliced context (Fig. 4b, c), and were not affected in the two other contexts. This suggests that SAFB is involved in preventing the nuclear export of unspliced RNAs. Analysis of ENCODE data showed that transcripts associated with SAFB eCLIP clusters were more nuclear in both spliced and unspliced forms ($P = 0.015$ for unspliced and $P = 1.36 \times 10^{-12}$ for the spliced form, Wilcoxon two-sided rank-sum test), with much larger effect sizes in the unspliced transcripts—transcripts from unspliced genes with >=2 SAFB eCLIP clusters in HepG2 cells were 2.8-fold more nuclear than those without SAFB eCLIP clusters, compared to just 1.47-fold for spliced genes (Fig. 4d).

**IGF2BP1 regulates RNA stability specifically in the context of linear spliced RNAs.** The insulin-like growth factor-2 mRNA-binding proteins (IGF2BP, also called IGF-II mRNA-binding protein (IMP), or Zipcode-binding protein (ZBP)) are a family of conserved RBPs that regulate a large repertoire of mRNA transcripts through diverse mechanisms[47] that are typically expressed only in embryonic tissues. IGF2BP proteins are upregulated in different types of cancers, and have a role in cell cycle progression in cancer cells[47]. IGF2BP1 and IGF2BP2 targets are highly overlapping, and both of them recognize the same CA-rich motif[48,49]. IGF2BP3, another member of the IGF2BP family, is not expressed in MCF-7 cells.

IGF2BP1/2 binding motif was enriched in cytoplasmic tiles in both spliced and unspliced libraries (Fig. 5a), but the localization of these tiles was more affected by the KD of IGF2BP1 (Fig. 5b)

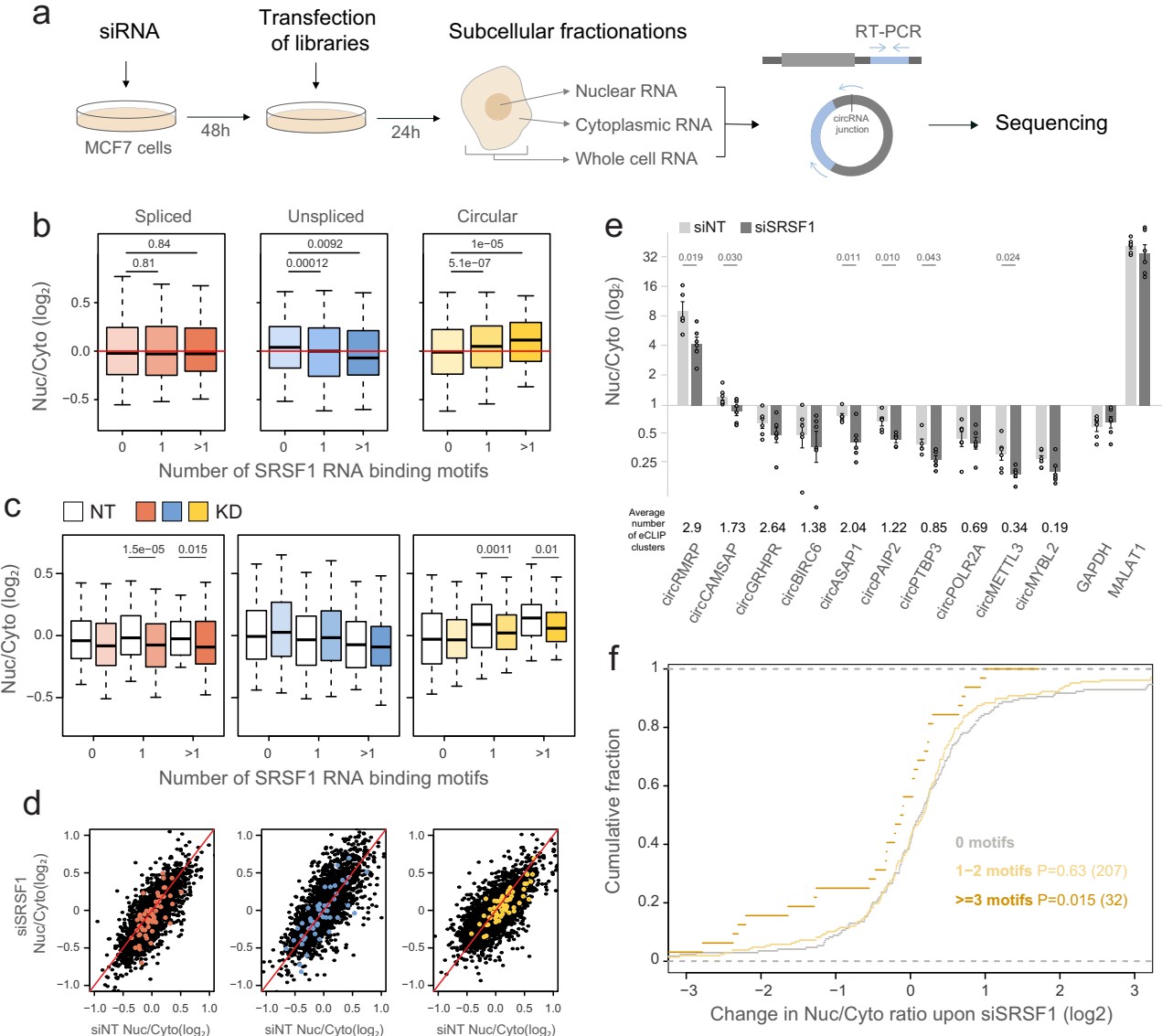

**Fig. 3 The effect of SRSF1 KD. a** Outline of the experimental procedure; cells were transfected first with a pool of siRNAs. After 48 they were transfected with NucLibA libraries, and were fractionated after an additional 24 h into nuclear and cytoplasmic fractions. Libraries for sequencing were generated from total, nuclear, and cytoplasmic RNA. **b** Nuc/Cyto ratios of tiles with the indicated number of SRSF1 motifs in each context; spliced (red), unspliced (blue), and circular (yellow). $n = 3$ biologically independent samples. Box plots show median, first to third quartile, whiskers are 1.5× interquartile range. $P$-values were computed using two-sided Wilcoxon rank-sum test. **c** Nuc/Cyto ratio of tiles with the indicated number of SRSF1 motifs following transfection of NT control (white) and siSRSF1 (colored as in **b**). $n = 3$ biologically independent samples. Box plots are as in (**b**). $P$-values were computed using two-sided Wilcoxon rank-sum test. **d** The effect of factors depletion on subcellular localization; Nuc/Cyto ratio of tiles in SRSF1 KD samples vs. NT control samples of all tiles (black), and tiles that have >1 motif (colored). Red line indicates X = Y. **e** qPCR analysis of nuclear and cytoplasmic fractions following transfection of siNT and siSRSF1 $n = 6$ biologically independent samples. Data are presented as mean values +/− SEM. The average number of SRSF1 eCLIP clusters across replicates in HepG2 cells is shown below each circRNA. **f** Change in Nuc/Cyto ratio for circRNAs with the indicated number of SRSF1 binding motifs. $P$-values are for comparing the indicated group with circRNAs with no SRSF1 motifs and were computed using a two-sided Wilcoxon rank-sum test. Number of circRNAs in each group is indicated in parentheses. Source data are provided as a Source data file.

compared to KD of IGF2BP2 (Supplementary Fig. 8a). There was also a significant correlation between increased nuclear presence of the tiles and increased expression levels upon KD, which was stronger for IGF2BP1 KD (Fig. 5c) compared to KD of IGF2BP2 (Supplementary Fig. 8b) and was particularly apparent in the eCLIP targets of IGF2BP1. IGF2BP1 KD affected Nuc/Cyto ratios of eCLIP targets in the spliced context by both increasing the levels of the enriched transcripts in the nucleus and decreasing them (to a lesser extent) in the cytoplasm (Figs. 5d and S8c). Since IGF2BPs have reported roles in RNA stability[47,50,51], we

examined the association between IGF2BP1 binding and RNA levels and the stability of tiles. Tiles with the IGF2BP binding motif were significantly less stable in the spliced context compared to the rest of the tiles ($P = 2.2 \times 10^{-16}$ for tiles that have >1 IGF2BP motif vs. all other tiles, Wilcoxon two-sided rank-sum test) (Fig. 5e). This suggests a role for IGF2BP1 in both nuclear export and destabilizing RNA, which leads to a shorter half-lives of the RNAs bound by it, and an increase in their expression, driven by nuclear accumulation, upon IGF2BP1 KD. The changes in overall expression and in localization were highly

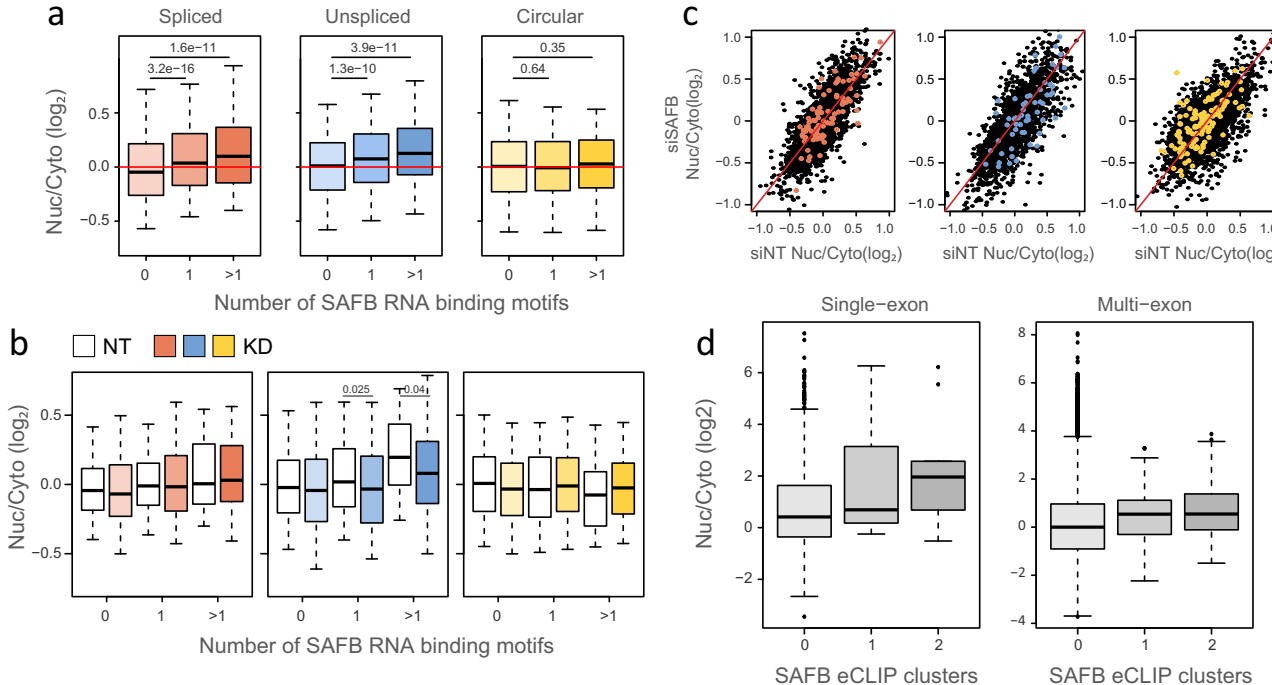

**Fig. 4 The effect of SAFB KD. a** Nuc/Cyto ratios of tiles with the indicated number of SAFB motifs in each context; spliced (red), unspliced (blue), and circular (yellow). $n = 3$ biologically independent samples. Box plots show median, first to third quartile, whiskers are 1.5× interquartile range. $P$-values were computed using two-sided Wilcoxon rank-sum test. **b** Nuc/Cyto ratio of tiles with the indicated number SAFB motifs following transfection of NT control (white) and siSAFB (colored as in **a**). $n = 3$ biologically independent samples. Box plots are as in (**a**). $P$-values were computed using two-sided Wilcoxon rank-sum test. **c** The effect of factors depletion on subcellular localization; Nuc/Cyto ratio of tiles in SAFB KD samples vs. NT control samples of all tiles (black), and tiles that have >1 motif (colored). Red line indicates X = Y. **d** Subcellular localization of single-exon and multi-exon transcripts annotated in RefSeq, harboring 0, 1 or more than 1 SAFB eCLIP clusters in ENCODE project RNA-seq data from HepG2 cells. $n = 1004$, 7, and 9 for single-exon and $n = 42002$, 134, and 185 for multi-exon transcripts. Source data are provided as a Source data file.

consistent, and the tiles that became more nuclear upon IGF2BP1 KD were the ones that were least stable in unperturbed cells (Fig. 5f), suggesting that the two processes are related.

In order to study the effects of IGF2BP1/2 on expression levels and subcellular distribution of endogenous long RNAs, we sequenced polyadenylated RNAs in the nucleus, cytoplasm, and whole-cell extracts of three replicates of control and IGF2BP1 or IGF2BP2 KD cells. Nuc/Cyto ratios were concordant between replicates ($R = 0.6–0.79$ between replicates). IGF2BP1 KD led to an increase in the nuclear enrichment of RNAs containing increasing numbers of IGF2BP1 eCLIP binding clusters (Supplementary Fig. 8d) and a similar trend was observed when considering RNA expression (Supplementary Fig. 8e), closely matching the observations using our reporters. Notably, changes were typically modest— transcripts from 160 genes became >2-fold more nuclear upon IGF2BP1 knockdown, a number much lower compared to the effects of knockdown of general export factors in MCF-7 cells[25]. A large number of IGF2BP1 binding events and nuclear enrichment upon IGF2BP1 depletion were mostly observed in multi-exon RNAs (Supplementary Fig. 8f), supporting its synergistic effect with splicing on RNA nuclear export. In the bound RNAs, an increasing number of binding clusters was associated with an increase in the expression levels in the nucleus and a decrease in the cytoplasm (Supplementary Fig. 8g), consistent with an effect on export rather than just on cytoplasmic stability. Depletion of IGF2BP2 had a different effect on the endogenous RNAs—transcripts with many IGF2BP2 binding sites were strongly reduced in their expression in the nucleus and to a somewhat lesser extent in the cytoplasm (Supplementary Fig. 8h), which resulted in an overall cytoplasmic shift in localization (Supplementary Fig. 8i).

In order to visualize the effect of IGF2BP1 on RNA export, we established lines of MCF-7 cells stably expressing a Doxycycline-inducible spliced β-globin mRNA that is fused to one of four different tiles (Mlxipl #73, NEAT1 #17, NEAT1 #81, and Hoxa11-AS#2). We also attempted to establish similar lines for the unspliced β-globin, but its expression levels in induced cells were too low to enable robust quantification. The tiles were selected based on the change in their localization specifically in the spliced and not in the unspliced context (Supplementary Fig. 9a), and on the presence of IGF2BP1 binding motifs and/or CLIP clusters (Supplementary Fig. 9b). We then used single-molecule fluorescence in situ hybridization (smFISH)[52] to visualize the localization of β-globin mRNA with each of the tiles or with an 'empty' 3′ UTR in cells treated with control or IGF2BP1-targeting siRNAs. For the Mlxipl #73 tile, knockdown of IGF2BP1 resulted in a strong reduction in expression levels in both the nucleus and cytoplasm, which prohibited robust estimation of Nuc/Cyto ratios. For the two tiles derived from NEAT1, we found that knockdown of IGF2BP1 resulted in a significantly more nuclear localization of β-globin compared to that of β-globin with an empty 3′UTR (Fig. 5g, h and Supplementary Fig. 9c, d). For Hoxa11-AS #2, adding the tile significantly enhanced cytoplasmic localization in control cells, but this enhancement was attenuated upon IGF2BP1 KD (Fig. 5g, Supplementary Fig. 9c, e). These results suggest that our observations on the positive effects of IGF2BP1 on the export of spliced RNAs extend to another expression system (inducible expression from a reporter integrated by viral infection) and to an imaging-based localization quantification method.

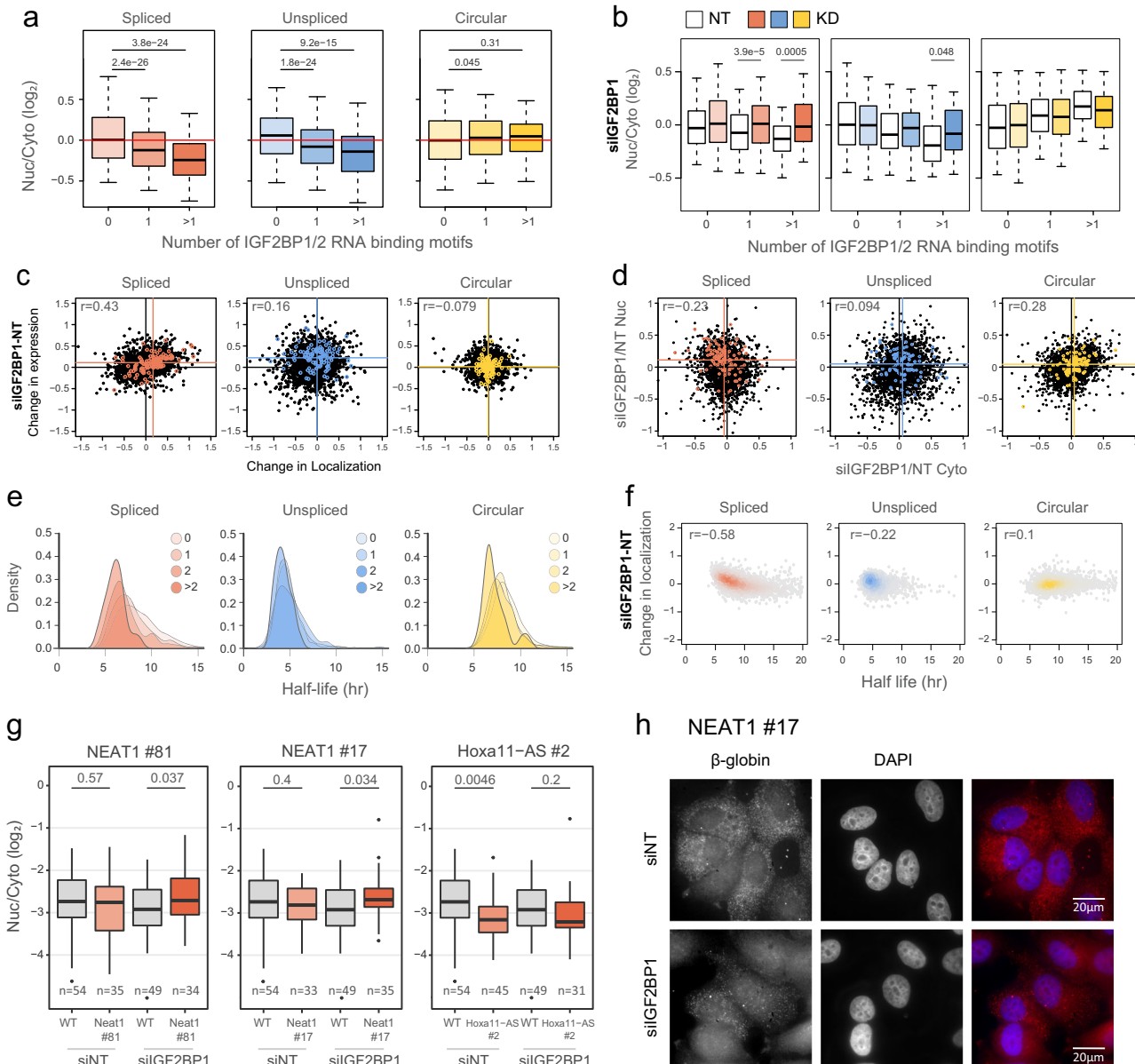

**Fig. 5 IGF2BP1 regulates nuclear export of linear spliced RNAs. a** Nuc/Cyto ratios of tiles with the indicated number of IGF2BP1/2 motifs in each context; spliced (red), unspliced (blue), and circular (yellow). $n = 3$ biologically independent samples. Box plots show median, first to third quartile, whiskers are 1.5× interquartile range. *P*-values were computed using two-sided Wilcoxon rank-sum test. **b** Nuc/Cyto ratio of tiles with the indicated number of IGF2BP1/2 motifs following transfection of NT control (white) and siIGF2BP1 (colored). $n = 3$ biologically independent samples. Box plots are as in (**a**). *P*-values were computed using two-sided Wilcoxon rank-sum test. **c** Correlation between the change in localization (X axis, normalized $\log_2$(Nuc/Cyto) values in KDs samples vs. control) and expression (Y axis, normalized WCE/input values in KDs samples vs. control) of all tiles (black) and tiles that have more than 1 eCLIP cluster (colored) upon depletion of IGF2BP1. Lines indicate X = 0 and Y = 0 (Black) and mean of all tiles that have an IGF2BP1 eCLIP cluster (colored). **d** Correlation between the change in expression in the cytoplasmic fraction (X axis, normalized values in KDs samples vs. control) and in the nuclear fraction (Y axis, normalized values in KDs samples vs. control) of all tiles (black) and tiles that have an IGF2BP1 eCLIP cluster (colored). Lines indicate X = 0 and Y = 0 (Black) and mean of all tiles that have an IGF2BP1 eCLIP cluster (colored). **e** Half-life distributions of tiles with the indicated number of IGF2BP motifs in each context. **f** The correlation between the change in localization (normalized $\log_2$(Nuc/Cyto) values in KDs samples vs. control) and half-life of tiles in each context. **g** Quantification of Nuc/Cyto ratios of β-globin signal as measured by smFISH of the inserted tiles relative to the WT β-globin sequence. $n \geq 31$ cells examined over 1 experiment. Box plots are as in (**a**). **h** Representative smFISH images of β-globin-NEAT1#17 (red) and DAPI (blue) in control and IGF2BP1-depleted cells. Source data are provided as a Source data file.

## Discussion

MPRNAs were recently used for the identification of RNA elements driving nuclear enrichment in different settings[22–25], and others have employed similar reporters to study the effects of computationally predicted elements[53]. While some motifs, including a C-rich motif driving nuclear enrichment, were reported in multiple studies[22,23], there have also been notable discrepancies. For example, elements from human *XIST* driving strong nuclear enrichment in one context[23] did not show nuclear enrichment activity when studied in our NucLibA-based study[22]. One possible reason for this difference is the use of tiles of different lengths in different studies, as some known localization elements are rather long[28,31], and so subsequences of different sizes may vary in their ability to recapitulate the activity of the

full-length sequence. As we show here, the form of the host RNA can also substantially influence the effect of short RNA fragments on subcellular localization of the whole RNA. The mechanism of this importance of context remains unclear and will likely be elucidated in future studies. We recently showed that unspliced RNAs exhibit drastically different dependencies on canonical mRNA export factors for their nuclear exit[25], as splicing enables a 'bypass' of the requirement for folded G/C-rich sequences enabling export of unspliced RNAs, likely though direct or indirect recruitment of NXF1. This may explain the differences we observe in contribution of individual sequence elements to export of spliced or unspliced RNAs, as sequences that facilitate efficient NXF1 recruitment may have a larger influence when placed in unspliced RNAs and only a marginal contribution in spliced ones. For the differences between linear and circular form, one possibility is that the export of linear RNAs is heavily influenced by proteins binding to the 5′ cap and to the poly(A) tail, which are absent from circular RNAs, which may have to recruit such factors indirectly. Another aspect to consider is that circular RNAs can adopt more compact folds compared to linear forms of the same sequence, as was recently demonstrated[54], which can influence the ability of RBPs to access and bind specific sequences, and thus influence RNA export.

In this study, we used backbones based on three sequences— the β-globin, circPVT1, and a scrambled sequence of circPVT1. As our focus was on effects of sequences on localization rather than on circularization, in the circPVT1 and SCRcircPVT1 backbones we inserted the tiles into the center of the circRNA, minimizing the effects of the variable sequence on circularization. We further used an exon-junction-spanning primer for library generation, which allowed us to ignore sequence changes that would affect localization and/or stability by affecting back-splicing efficiency. We note that the high correlation between circPVT1 and the SCRcircPVT1 backbones, and the lack of similarity to the linear version of circPVT1, all suggest that the backbone sequence has an overall limited effect on the activity of sequence elements that influence localization and/or stability.

While it is now well appreciated that a large variety of circRNAs are produced in mammalian cells, their biology remains poorly understood. Much emphasis has been put on studying factors contributing to circRNA biogenesis and their potential translation[9], whereas regulation of circRNA subcellular localization or stability have not been extensively studied. The post-transcriptional fate of circRNAs is particularly interesting, as the main known pathways for export and decay of long RNA heavily rely on the 5′ cap and the poly(A) tail, which are absent in circRNAs. The paucity of these access points for decapping and deadenylation are thought to underpin the relative stability of circRNAs, and are likely a key feature in the functionality of some circRNAs, as they can allow circRNAs to remain stable while bound to RBPs that typically trigger decay though exonucleolytic activities. Here we report that cytoplasmic localization and stability of circRNAs are also correlated with each other, albeit weakly, supporting the notion that long-lived RNAs accumulate in the cytoplasm, which they likely reach when the nucleus disassembles during mitosis. Nevertheless, the variability we observe in both stabilities and Nuc/Cyto ratios of different circRNAs suggest that specific pathways, such as SRSF1 binding, sculpt the specific post-transcriptional fate of individual circRNAs.

We link the activities of three RBPs, SRSF1, SAFB, and IGF2BP1, to the fates of different subsets of RNA molecules. We note that since we followed up on just four factors in this study, and ENCODE eCLIP analysis suggests many additional factors with enrichment patterns at least as strong as the factors we studied (Supplementary Fig. 8a), it is likely that many additional factors sculpt the context-specific effects on localization. Our observations relate to previous studies of the factors we focused on. It was shown that SR proteins are recruited to the processed RNA together with the assembly of the spliceosome[55,56], and that exonic splicing enhancers are associated with efficient NXF1-dependent export[25], although in those studies SR proteins had a positive effect on export, whereas for SRSF1 we observe a negative effect. Depletion of SRSF1/SF2 ortholog in *D. melanogaster*, was shown to lead to accumulation of various circRNAs in fly cells[29,57]. It has been suggested that SRSF1 primarily inhibits circRNA biogenesis, but our results suggest that it also inhibits circRNA export, which may also limit circRNA accumulation. Indeed, SRSF1 KD resulted in increased expression of several circRNAs (Fig. 3e), and the decrease in Nuc/Cyto ratios was significantly correlated with increase in expression levels in our MPRNA experiments (Supplementary Fig. 4d). In contrast to SRSF1, which preferentially retains spliced RNAs in the nucleus, SAFB preferentially inhibits export of unspliced RNAs. Interestingly, SAFB binds LINE elements and was one of the top hits in a screen designed to identify proteins that are required for preventing L1 retrotransposition[58–60]. Given our results, an attractive hypothesis is that SAFB binding serves to inhibit nuclear export of the retrotransposon RNAs. Interestingly, two SAFB-related proteins, SAFB2 and SLTM were recently implicated in nuclear enrichment of long RNAs[61,62], suggesting that nuclear enrichment of bound RNAs might be a broad role of the SAFB family.

IGF2BP (IMP/ZBP) factors have been extensively studied in the context of RNA stability and localization[47], with the majority of studies focused on their roles in the regulation of specific genes, such as *MYC* and β-actin. IGF2BP1 is localized in both the nucleus and the cytoplasm in human cell lines (http://rnabiology.ircm.qc.ca/RBPImage/), and was shown to associate with its canonical target, β-actin mRNA, in the nucleus[63], consistent with a broader role in RNA export. There is also evidence that it may contribute to the export of short non-coding RNAs[64]. IGF2BP proteins were previously shown to have highly overlapping target sites[49,65], and their combined or independent depletion led to a mild reduction in expression of their targets in HeLa and HepG2 cells[65,66], while IGF2BP1 depletion did not preferentially affect levels of the bound targets in human embryonic stem cells[49]. Despite the fact that these factors bind closely similar motifs, their KD has a divergent effect on the fate of their targets in MCF-7 cells. IGF2BP1 contributes to RNA export and to its destabilization in the cytoplasm, as its KD leads to both nuclear retention and an increase in the expression of the targets. In contrast to IGF2BP1, IGF2BP2 KD led to a reduction in target expression levels, in both the nuclear and the cytoplasmic fractions. In both cases, the effects are strongest for spliced RNAs, and no strong effects are observed in a circular context.

One limitation of our study is that the sizes of the effects on Nuc/Cyto ratios elicited by the individual tiles are typically quite modest and rarely exceed 2-fold. Such effects are comparable to those observed in other studies using a similar system[22,23,62]. These limited effects could stem from the limited potency of individual short sequences that typically act in the context of much longer RNAs, and it is unclear how individual short sequences synergize together in the context of long RNAs. For example, the sequences we identified as influential when using the β-globin backbone may synergize in specific ways with the known elements in β-globin that inhibit export[28]. It is also possible that imperfections in the nuclear/cytoplasmic fractionation procedure, where some residual cytoplasmic RNA occasionally remains associated with the nuclei, limit the ability to detect large effect sizes. Notably, a series of recent MPRNAs applied to neurite/soma fractionations, some using the same experimental design as we do here, occasionally found much larger effect sizes, even

though in many cases no short elements could be found to recapitulate the activities of full-length 3′UTRs[67–69].

Beyond the RNA form, we expect that additional features will have important roles in influencing how a specific short sequence affects RNA fate. The transcriptional context of the host RNA, and specifically whether the elements are transcribed from chromosomes or from episomal vectors can play a substantial role, due to the tight coupling between transcription and export[70]. The Wilusz laboratory has shown that circRNA size is important for their export pathway and efficiency[16]. In our setting, we studied circRNAs of a fixed length of 560–590 nt, corresponding to the 'intermediate' length range in ref. [16], where circRNA export can be regulated by either UAP56 or URH49. We have previously found that RNA length is also associated with the export efficiency of linear RNAs[71], and showed that the export of unspliced or long-exon-bearing RNAs is NXF1-dependent and mediated by structured and G/C-rich regions[25]. Relatedly, Kudla and colleagues have observed a strong dependence on G/C-content of the export of unspliced but not spliced RNAs[72], findings which also have support in studies of individual genes[73]. Future studies will elucidate how transcription efficacy and chromatin status, splicing, polyadenylation, RNA size, RNA modifications, and G/C context, all synergize with specific sequence elements recruiting RBPs to sculpt RNA fate.

## Methods

**Cell culture**. MCF-7 (ATCC) were maintained in DMEM (Gibco) medium, supplemented with 10% Fetal Bovine Serum (Gibco) and 1% Pen-Strep (Biological Industries), in a humidified 5% $CO_2$ incubator at 37 °C. Cells were routinely tested for mycoplasma contamination using Mycoplasma test kit (Hy-KPC detection, HyLabs).

**Library design**. Oligonucleotide pools were designed as in ref. [22] and synthesized by Twist Bioscience (San Francisco). Backbone sequences and insertion positions are given in Supplementary Data 5.

**RNase R treatment followed by polyadenylation and poly(A) + RNA depletion (RPAD)**. RPAD was performed according to the protocol as described in ref. [26]. In brief, 10 μg of RNA from nuclear and cytoplasmic RNA was incubated with RNase R (ABM, E049) for 30 min at 37 °C. Treated RNA was isolated using miRNeasy Mini Kit (Qiangen) following the manufacturer's instructions and eluted with 30 μl of nuclease-free water. RNA was then incubated with polyA polymerase (NEB, M0276) for 30 min at 37 °C. 100 μl of Oligo-d(T)25 Magnetic Beads (NEB, S1419) were washed one time with lysis/binding buffer according to manufacturer's instructions and were suspended in X2 lysis/binding buffer. RNA was incubated at 70 °C for 5 min, and was then incubated with the washed beads. Samples were incubated with shaking for 20 min. RNA samples with Oligo-dT beads were placed on a magnetic stand and the supernatant was collected. RNA was isolated using miRNeasy Mini Kit. Libraries for sequencing were generated using SENSE Total RNA-Seq Library Prep Kit (Lexogen, LX-042).

**circRNA quantification in RPAD libraries**. RNA-seq reads were mapped to the genome with STAR with the following parameters: "–outSAMstrandField intronMotif –outSAMtype BAM SortedByCoordinate –outWigType bedGraph –chimOutType SeparateSAMold –chimSegmentMin 15 –alignSJoverhangMin 15 –alignSJDBoverhangMin 15 –chimScoreSeparation 10 –chimJunction OverhangMin 5". The chimeric read pairs identified by STAR were parsed and compared with circRNA boundaries annotated in CircBase and in publications focused on circRNAs in breast cancer cells and in other cell types[8,74,75]. Chimeric junctions within 5 nt of the annotated circRNA boundaries of each annotated circRNA were counted and used to compute Cyto/Nuc ratios.

**CircLibA and NucLibA libraries plasmid construction**. Cloning of CircLibA and NucLibA into expression vectors was performed using restriction-free (RF) cloning[76]; Oligonucleotide pool was amplified by PCR using Q5 High-Fidelity DNA Polymerase (NEB, M0491), and purified using AMpure beads (A63881, Beckman) at a 2:1 beads:sample ratio according to the manufacturer's protocol. For adding flanking regions of target plasmid, PCR product was amplified using primers containing parts of the β-globin, circPVT1, or SCRcircPVT1 constructs (Supplementary Data 2). Products were concentrated using Amicon tubes (UFC503096, Millipore) and purified using AMpure at a 2:1 beads:sample ratio. PCR product was used in a PCR with WT β-globin, Δintrons β-globin, circPVT1 or SCRcircPVT1 construct and X1 KAPA HiFi HotStart ReadyMixPCR (Kapa Biosciences). Reactions were then treated with 10 μl DpnI for 1 h at 37 °C to digest the

methylated parental plasmid, and purified using AMpure beads at a 1:1 beads:sample ratio. RF product was transformed into Escherichia coli electrocompetent bacteria (60117-2, Lucigen), and plated on 12 × 15-cm LB/Amp agar plates. Colonies were scraped off the plates and DNA was extracted using a plasmid midiprep kit (K210014, Invitrogen).

**ENCODE RNA-seq data analysis**. RNA-seq libraries from nuclear and cytoplasmic fractions of HepG2 cells from the ENCODE project were quantified using human RefSeq annotations using RSEM[77] and intersected with ENCODE eCLIP clusters as in ref. [22].

**Plasmids transfection**. Twenty-four hours prior to transfection, $10^6$ cells were seeded in a 10 cm plate (Corning). Plasmid transfections were performed using PolyEthylene Imine (PEI) (PEI linear, Mr 25,000, Polyscience)[78].

**Extraction of cytoplasmic and nuclear RNA**. Cells were washed in cold PBS and detached from plates by 10 mM EDTA, a fraction was transferred to a new tube and RNA was extracted using TRIREAGENT (TR 118, MRC) to obtain whole-cell extract (WCE). Remaining cells were washed in cold PBS, resuspended in 150 ml RLN buffer (50 mM TrisHCl pH = 8, 140 mM NaCl, 1.5 mM $MgCl_2$, 10 mM EDTA, 1 mM DTT, 0.5% NP-40, 10 U/ml RNase inhibitor), and incubated on ice for 5 min. The extract was centrifuged for 5 min at 300 × g in a cold centrifuge, and the supernatant was transferred to a new tube and centrifuged again for 1 min at 500 × g. The supernatant (cytoplasmic fraction) was transferred to a new tube and RNA was extracted using TRIREAGENT. The nuclear pellet was washed once in 150 ml RLN buffer, resuspended in 1 ml of buffer S1 (250 mM Sucrose, 10 mM $MgCl_2$, 10 U/ml RNase inhibitor), layered over 3 ml of buffer S3 (880 mM Sucrose, 0.5 mM $MgCl_2$, 10 U/ml RNase inhibitor), and centrifuged for 10 min at 2800 × g in a cold centrifuge. The supernatant was removed and RNA was extracted from the nuclear pellet using TRIREAGENT.

**siRNA transfections**. MCF-7 cells were transfected with 10 nM siRNA pool (Supplementary Data 2) or non-targeting control using Lipofectamine 3000 reagent (Invitrogen) according to manufacturer's protocol. Forty-eight hours after transfection, cells were washed and transfected with CircLibA or NucLibA plasmid pool. After an additional 24 h, cells were fractionated as described above, and RNA was extracted from cytoplasmic, nuclear, and WCE.

**qPCR analysis**. For analysis of gene expression and RNA subcellular localization, RNA was isolated from culture cells with TRIREAGENT followed by DNase treatment (Perfecta DNase, Quanta Biosciences). cDNA was synthesized using 1 μg of RNA (measured by Nano-drop spectrophotometer), with qScript Flex cDNA kit (Quanta Biosciences). Quantitative real-time PCR (qRT-PCR) was performed using Fast SYBR qPCR mix (Thermo Fisher) and analyzed on ViiA7 real-time PCR machine (Applied Biosystems) in a final reaction volume of 10 μl. The primer sets used for qRT-PCR are listed in Supplementary Data 3. For differential expression analysis in WCE RNA samples, expression levels were normalized to GAPDH, and to their relevant control (ΔΔCt method). Nuc/Cyto ratios were computed without normalization using ΔCt values, fractionation quality was validated by primers targeting the nuclear MALAT1 gene, and the cytoplasmic GAPDH mRNAs.

**Sequencing and amplicons library generation**. One microgram of RNA was used for cDNA production using the qScript Flex cDNA synthesis kit (95049, Quanta) and a gene-specific primer containing library 3′ adapter and part of the Illumina RD2 region. The entire cDNA reaction was diluted into 100 mL second strand reaction with a primer containing a unique molecular identifier (UMI), library 5′ adapter and part of the Illumina RD1 region. The second strand reaction was carried for a single cycle using Phusion Hot Start Flex DNA Polymerase (NEB, M0535), purified using AMpure beads at a 1.2:1 beads:sample ratio and eluted in ddH2O. Product was used for amplification with barcoded primers, and the amplified libraries were purified by two-sided AMpure purification; First with a 0.5:1 beads:sample ratio followed by a 0.7:1 ratio. Libraries were sequenced a NextSeq 500 or NovaSeq 6000 machine to obtain 150 nt single-end reads.

**Transcription inhibition by actinomycin D and half-life calculation**. MCF-7 cells were transfected with plasmid libraries as described above. Twenty-four hours post-transfection, growth medium was replaced and actinomycin D (A9415, Sigma) was added to a final concentration of 5 μg/ml. DMSO (MP Biomedicals,1:400) was used as control. Cells were collected after 2, 4, and 8 h of actinomycin D or DMSO treatment using TRIREAGENT spiked with 2 RNA oligos transcribed in vitro from a DNA template encoding Renilla Luciferase at concentrations of 0.002 ng/ml and 0.0002 ng/ml. Libraries were prepared as described above, and expression levels of the tiles were normalized to luciferase levels.

Half-life was calculated by first fitting a linear regression model between the time points and the expression levels. The intercept and slope were extracted from

the regression equation. The half-life was then calculated as (logb(0.5)-lm_intercept)/lm_slope.

**In vitro transcription (IVT)**. Templates for in vitro transcription were generated by amplifying 2 regions of Renilla Luciferase sequence from a plasmid using Phusion Hot Start Flex DNA Polymerase (NEB, M0535), with primers that contain the adapters of NucLibA and CircLibA (Supplementary Data 2). PCR products were amplified by adding the T7 promoter to the forward primer. RNA was produced using MEGAscript T7 in vitro transcription kit (Ambion), and template DNA was removed by treating with DNaseI (Ambion).

**Library data analysis**. The sequenced reads were used to count individual library tiles as in ref. [22]. We considered only R1 reads that contained the TAG-GAGGCCTCATCTGACTG adapter sequence, and extracted the unique molecular identifier (UMI) sequence preceding the adapter. Each read was then matched to the sequences in the library, without allowing insertions or deletions. The matching allowed mismatches only at positions with Illumina sequencing quality of at least 35 and we allowed up to two mismatches in the first 15 nt ('seed'), and no more than four overall mismatches. If a read matched more than one library sequence, the sequence with the fewest mismatches was selected, and if the read matched more than one library sequence with the same number of mismatches, it was discarded. The output from this step was a table of counts of reads mapping to each library sequence. Only fragments with at least 20 reads were used in subsequent analysis. We then used these to compute nuclear/cytoplasmic and WCE/plasmid ratios after adding a pseudocount of 0.5. Tiles sequences, GC content, DeltaG, expression levels and Nuc/Cyto ratios in each context are listed in Supplementary Data 3.

**RNA-seq and data analysis**. WCE, cytosolic and nuclear fractions obtained after KD of IGF2BP1 and IGF2BP2 and fractionation were used to generate libraries using the SENSE mRNA-Seq Library prep kit (Lexogen) according to manufacturer's protocol and sequenced on a NovaSeq 6000 machine to obtain 100 nt paired-end reads. RNA-seq reads were mapped to the human genome (hg19 assembly) with STAR (Dobin et al. [79]) and GENCODE v26 annotations, and gene expression levels were quantified using RSEM[77]. To quantify subcellular localization, Nuc/Cyto ratios were calculated using RSEM output of genes with average FPKM > 1. Differential expression in total, Cyto and Nuc samples were computed using DESeq2 (Love et al. [80]) with default parameters. Genes parameters, expression levels in WCE, cytosol and nucleus and normalized Cyto/Nuc ratios are listed in Supplementary Data 4.

**eCLIP data analysis**. eCLIP clusters as defined by ENCODE were obtained from http://encodeproject.org. Only clusters with significance P < 0.01 and at least 2-fold enrichment over mock input control were considered. "bedtools intersect" was used in order to identify tiles overlapping eCLIP clusters.

**RNA secondary structure**. ΔG was computed using the program RNAfold of the ViennaRNA package with default parameters[81,82].

**Generation of stable lines for smFISH**. To generate cell lines with inducible expression of β-globin sequence variants, we first cloned WT β-blogin cDNA into the pRAR expression vector using restriction-free (RF) cloning[76], followed by cloning of the specific tiles from NucLibA. Lentiviral particles were generated as previously described[83]: HEK293T cells were transfected with a mixture of pRAR plasmid:pMDL:pVSVG:pRev at the ratio of 1:0.65:0.35:0.25, respectively, using PEI. Medium was collected from plates 48 h after transfection and filtered.

MCF-7 cells were infected by the addition of growth medium containing lentiviral particles and 8 µg/ml Polybren (Sigma). Selection was done by the addition of Puromycin (InvivoGen) at a concentration of 1 µg/ml. Pools of infected cells that survived the selection were used for smFISH; cells were transfected with siRNAs (see above), and 32 h after transfection Doxycycline (Sigma, D9891) was added to a final concentration of 1 µg/ml. Cells were grown for an additional 16 h and were fixed using 4% PFA in PBS.

**Single-molecule FISH (smFISH)**. Probe libraries were designed according to Stellaris guidelines and synthesized by Stellaris (Stellaris RNA FISH probes, Biosearch Technologies) as described in Raj et al.[52]. Library targeting β-blogin consisted of 22 probes labeled with Quasar 570 (Supplementary Table 2). Hybridization conditions and imaging were conducted as described previously[84]. Hybridizations were done overnight at 30 °C with probes at a final concentration of 0.1 ng/ml. Images were taken with a Nikon Eclipse Ti2 fluorescence microscope equipped with a X100 oil-immersion objective and a Photometrics Prime 95B camera, using NIS-Elements Advanced Research software. Quantification was done with FishQuant V3[85]. We performed automatic 2D projections as suggested in FishQuant documentation, followed by automatic cell segmentation using CellProfiler. DAPI signal was used to segment nuclei and β-blogin was used to segment cell bodies. Following batch analysis, we manually examined segmentation and removed incorrectly

segmented cells from further analysis using Fiji (ImageJ) software. Quantification of cytoplasmic and nuclear signals was performed with default parameters and recommended filters of FishQuant.

## Data availability

The data supporting the findings of this study are available from the corresponding authors upon reasonable request. Sequencing data generated in this project have been deposited in the SRA database under accession PRJNA705592.

## Code availability

Code for counting MPRNA tags and computing ratios is archived at https://zenodo.org/record/6352589[86] and the most recent code can be found https://github.com/igorulitsky/MPRNA.

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

## Acknowledgements

We thank Noam Stern-Ginossar, Schraga Schwarz, Yaron Shav-Tal, and members of the Ulitsky lab for helpful discussions and comments on the manuscript. Yoav Lubelsky for help with establishing the experimental assays. Liat Alyagor and Dana Hirsch for help with the

FISH assays. Tamar Gera for help with establishing the RNA stability assays. Jeremy Wilusz for the ZKSCAN1 expression vectors and useful discussions on the minimal size of circularizable RNA. Joan Steitz for the β-globin plasmid. This work was supported by grants to I.U. from the Israel Science Foundation (852/19 and 2406/18) and The EU Joint Programme - Neurodegenerative Disease Research (JPND ERA-Net localMND).

## Author contributions

M.R. and I.U. conceived the study. M.R. performed all experiments and M.R. and I.U. analyzed the data. M.R. and I.U. wrote the manuscript.

## Competing interests

The authors declare no competing interests.

## Additional information

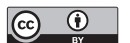

