## [Peer Review File · Nature Communications]

Title: Context-specific effects of sequence elements on subcellular localization of linear and circular RNAsREVIEWER COMMENTS

Reviewer #1 (Remarks to the Author):

The authors describe MPRA experiments showing that the RNA sequence code for nuclear localization of circular RNA vs. linear RNAs appear distinct. This is an interesting observation that should be reported, but several concerns diminish the strength of the results and consequent interpretations.

1. The effect size of circular RNA localization in the MPRA experiment appears quite low, especially compared to known mRNAs and lncRNAs. This is a concern because all of the MPRA screens are based on this background. With very small effect sizes, what is the actual impact on circRNA function? It is also a surprise that all of the localization data are based on fractionation; more in situ imaging would be helpful.
2. The authors make an interesting discovery that the circPVT1 exhibits similar localization patterns to “scrambled” SCR circPVT1. They so strongly believe in this correlation that they cite this as a reason to only test non-scrambled circPVT1 under the results section “Context-specific effects of sequence elements on RNA stability.” This correlation seems to suggest that the localization mechanism for circular RNAs may be heavily driven by the circular structure, and less so by the specific sequence motifs. However, the authors then proceed to claim that circular RNA localization is heavily driven by sequence, for example when discussing SRSF1. I would like to see the authors explore the role of sequence versus structure more, and more explicitly acknowledge the tension or balance between these two ideas.
3. In fact, according to S5A, SRSF1 drives greater nuclear enrichment in the scrambled circRNAs. Ostensibly, scrambled circRNAs should carry few motifs, if any. How can this result be reconciled?
4. For the results surrounding RBPs, the authors specifically discuss SRSF1, SAFB, IGF2BP1, and IBF2BP2. Firstly, IGF2BP2 is not shown in S5A. Furthermore, how were these selected? The text states that they were chosen based on enrichment – I would like to see more specificity around this statement. Are these the genes with the largest fold change difference? Most significant? Largest difference and are recognizable genes? Without such details, these results feel somewhat cherry picked.

Minor points:

- For determining significant enrichment, the authors use a cutoff of 0.3 log fold change; please provide justification for this value.
- The authors make a point to compare G/C content of transcripts localizing to different regions. However, it is unclear how the p-values attached to these comparisons are generated. Is this a t-test? Mann-Whitney test? The Wilcoxon rank-sum test is used in other scenarios, which leads me to believe this may be test used here as well, but this is not clearly discussed.
 - o Similarly, for the ΔG calculations for Figure 2D. Additionally, if the authors wish to cite the “spliced” context for the structured-ness, please provide a p-value for that as well.
- How is ΔG calculated? There may be “standard” method I’m not aware of, but I couldn’t seem to find a detailed description of this method.
- Axes for Figure S4D are somewhat confusing – I would intuitively expect that GC content would be on the x-axis, as half life is ostensibly a function of GC content, not the other way around.

o Similar for Figure 4F

- It appears that higher GC content actually leads to more stability in unspliced transcripts in Figure S4D; I think the authors should mention this, as this is a more similar match to prior results from Litterman et al.

- Across several figure panels, the authors vacillate between showing p-values, or using asterisks to indicate significance. Recommend showing p-values for consistency and transparency.

- Figures 4G, H, and I are confusing. For example, for 4G, it is described as showing change in localization of genes given different numbers of eCLIP clusters. However, the plot itself does not seem to really show the “baseline” unless that baseline is 0 (unclear). Additionally, the statistical testing is unclear to me here. The asterisks seem to indicate significant difference between a group and the union of all other groups that are either lower or higher. This seems like a strange way to do statistical testing. If the takeaway here is that increasing eCLIP clusters increases effect of knockdown, wouldn't it be more direct to use something akin to a Spearman's correlation? Or, simply bisect the data by number of eCLIP clusters and compare high versus low? Even if the authors insist on this type of comparison, it seems prudent here to do some sort of multiple hypothesis correction here (which is not mentioned), seeing as they are essentially comparing “gain” and “losses” across 6 different comparisons.

Reviewer #2 (Remarks to the Author):

In this manuscript, Ron and Ulitsky use MPRNA to conduct a broad survey of cis-elements that promote or inhibit the nuclear export of unspliced, spliced and circular reporter RNAs. They find that elements with high GC content promote export of unspliced and spliced RNAs, similar to their previous findings and from those from the Kudla group. Interestingly, this does not appear to affect circular RNAs. Based on these results, they then go on to test whether trans-elements (RNA binding proteins), whose motifs are enriched in nuclear or cytoplasmic RNA pools, affect nuclear export or nuclear retention in MPRNA experiments. From this they find that distinct types of RNA (i.e. unspliced, spliced and circular RNAs) are differentially responsive to different trans-factors. In particular SRSF1 appears to promote the nuclear retention of spliced and circular RNAs, SAFB promotes the nuclear retention of unspliced RNAs, and IGF2BP1 promotes the export of spliced and unspliced RNAs. They then examine how IGF2BP1 and 2 affect the nuclear/cytoplasmic distribution of endogenous mRNAs.

The paper is very interesting and of broad interest to the mRNA community. I am generally in favor of publication as long as some of the following two major concerns are addressed.

- 1) Much of the paper is based on RNAseq analysis. The results should be validated by imaging data – this should not be too hard as examples of reporters with elements that promote nuclear retention or export can be individually expressed and visualized by fluorescent in situ hybridization (FISH). The authors should provide examples of cis-elements that promote retention, and export. The authors should also validate that the depletion of trans-factors (SRSF1, SAFB and IGF2BP1) alters these distributions. Finally, authors should validate that depletion of IGF2BP1 affects the nuclear/cytoplasmic

distribution of particular endogenous poly(A)-mRNAs (as predicted by their analysis) by single molecule FISH.

2) The effect of IGF2BP1/2-depletion on endogenous mRNAs does not seem very significant. How does this compare to depletion of UAP56 or NXF1? Moreover, information on the experiment is inadequate. Was the RNAseq data from multiple independent biological replicates? Are data from replicates consistent? For details, the text refers the reader to Table S4 (I presume that this is “S4 Supplementary Dataset”) but the data presentation in that file is opaque. The enhanced effect on the cytoplasmic/nuclear ratio of mRNAs with many eCLIP clusters (Figure 4G) could simply be an effect on long mRNAs rather than short mRNAs.

Minor concerns:

- Intro: The authors should include details about circRNA conservation and estimates of function (PMID: 34320353). The idea that circRNAs may simply be byproducts that accumulate as they do not cause too much harm, but could eventually be disruptive is in line with the association of circRNAs with aging and neurodegeneration (PMIDs: 29753875, 34016449).
- Figure S1E: Specify in the figure legend what RT “+” and “-” mean (reverse transcriptase followed by PCR, vs DNA prep followed by PCR?)
- Figure S1 and S7: The size of the nucleotide standards should be labelled.
- Figure 2D: Although there is a correlation between “folding” (i.e., delta-G) and export, this is what you would expect given the GC content results. Is the amount of folding in export-promoting sequences greater than what one predicts given the correlation with GC content?
- Page 8: “information driving the strong nuclear enrichment of this lncRNA ... has adapted specifically to be effective within human cells” this is speculative – no data presented suggest that the nuclear-localization activity of MALAT1 fragments is a particular adaptation which has occurred in humans.
- Figure S3: Fragments from the 3’ end of MALAT1 appear to strongly inhibit export of the circRNAs – are these part of the masRNA region? Or part of the triple helix at the junction between MALAT1 and masRNA?
- Litterman 2019a and Litterman 2019b are the same publication.
- Page 12: “In contrast to a previous study, using a different reporter system (Litterman et al., 2019b), we found that higher G/C content was associated with reduced RNA stability, mostly in the spliced linear context.” The Litterman study reports that GC content in the 3’UTR correlates with a reduction in mRNA stability (the opposite of what was stated by the authors).
- siRNA sequences should be provided.
- The effects of SRSF1 on nuclear retention are surprising, although the data seems to indicate that the effect seems modest at best (Figure S5D). How does this compare to changes in export when nuclear export factors are depleted (e.g. in Zuckerman et al., 2020)?
- There are some inconsistencies – in Figure 3F the correlation between SAFB binding motifs and nuclear retention seems similar in spliced and unspliced, while in Figure 3G the white box plots in these two data sets look different. Could the lack in change in localization of spliced mRNAs for SAFB knockouts be due to inconsistencies in the control (NT) data set (comparing 3F and 3G)?
- Similar to the previous point, the correlation between IGF2BP1 binding motifs and nuclear localization

of circular RNAs for control-depleted cells looks different between Figure 4A and 4B (no correlation in 4A, nuclear retention in 4B).

- The axes should be as clear as possible to help the reader interpret the figures (i.e., “change in expression” and “change in localization” should be more specific – see figures 4C, 4F, and S5D).
- Page 20 “... more than 1 SRSF1 eCLIP clusters ...” this should be IGF2BP1.
- Page 20/21 “Change in localization of genes ...” this should be “Change in localization of RNAs from genes ...” – similarly “Change in localization of single-exon genes” should be “Change in localization of RNAs from single-exon genes”.
- Discussion: beta-globin has cis-elements that inhibit export (PMID: 26362019) could inserted elements disrupt or enhance these? Likewise Pvt1 appears to contain many elements that promote export (Figure S3A) - are these enhanced or disrupted by some of the inserts? The authors should discuss these points.
- Discussion: Many documented nuclear retention elements appear to be long (e.g. PMIDs: 22355166, 26362019) - are these composed of many smaller elements that act in an additive manner, or do these form larger active units that lose nuclear retention activity when subdivided? The authors should discuss this.

Reviewer #3 (Remarks to the Author):

In this work the authors used massively parallel RNA reporter assays to study how sequences derived from lncRNAs or circRNAs affect the subcellular localization and stability of linear and circular RNA transcripts. They also performed in-depth analyses of selected RBPs (SRSF1, SAFB, IGF2BP1/2) to study their effects on transcript localization and stability. Overall, this work generated a large amount of data, with some interesting and potentially important observations. However, the present manuscript has some limitations and minor issues that need to be addressed in a revision.

1. Conceptually, the most interesting observation from this work is that the same sequence elements could have distinct impacts on transcript localization and stability, depending on whether they are in linear or circular RNA transcripts. However, there is not much mechanistic investigation or discussion about this observation. At minimum, the authors should expand the Discussion section to offer a more detailed possible mechanistic interpretation of this finding. Also, to what extent can this difference be attributed to sequence difference between the backbone vectors used for the linear vs circular RNA reporter?

2. The present work is largely based on massively parallel RNA reporter assays. It would be helpful if the authors can perform more in-depth studies to see if and how patterns learned from the reporter assays can be translated into endogenous genes. There are some efforts to do this (e.g. sequencing and analysis of polyA⁺ RNAs upon IGF2BP1/2 knock-down) – but I think this kind of analysis should be expanded. For example, while the authors performed qPCR analysis of 10 selected circRNAs in nuclear and cytoplasmic fractions following SRSF1 knockdown (Fig. 3E), they should carry out a more comprehensive analysis by performing circRNA sequencing of these RNA samples.

3. The authors seemed to have used ENCODE RNA-seq data of subcellular fractions in various analyses (one example being Fig. 3I), but there is no description in the Methods section about where these data came from or how they were analyzed.
4. Fig. S6, panel B legend. 'SRSF1' seems to be a typo for 'IGF2BP2'.
5. Fig. S7 in the Discussion section describes primary data, thus should be moved to the Results section.
6. Methods section – can the authors elaborate how RNA stability was calculated from the actinomycin D treatment data?

Point-by-point response to reviewers for “Context-specific effects of sequence elements on subcellular localization of linear and circular RNAs” by Ron & Ulitsky

Reviewer 1

The authors describe MPRA experiments showing that the RNA sequence code for nuclear localization of circular RNA vs. linear RNAs appear distinct. This is an interesting observation that should be reported, but several concerns diminish the strength of the results and consequent interpretations.

1. The effect size of circular RNA localization in the MPRA experiment appears quite low, especially compared to known mRNAs and lncRNAs. This is a concern because all of the MPRA screens are based on this background. With very small effect sizes, what is the actual impact on circRNA function? It is also a surprise that all of the localization data are based on fractionation; more in situ imaging would be helpful.

We agree with the reviewer that the overall effects measured in the MPRA for localization (in both other studies and in those from our lab) are typically modest, mostly <2-fold. However, we note these are achieved with relatively short ~100 nt sequences, whereas in the endogenous context the activities of such sequences synergize in the context of longer RNAs to produce larger effect sizes that ultimately lead to the variety of subcellular localizations of both linear and circular RNAs. We now expand on this in the Discussion section (page 29):

“One limitation of our study is that the sizes of the effects on Nuc/Cyto ratios elicited by the individual tiles are typically quite modest and rarely exceed 2-fold. Such effects are comparable to those observed in other studies using a similar system^{22,23,62}. These limited effects could stem from the limited potency of individual short sequences that typically act in the context of much longer RNAs, and it is unclear how individual short sequences synergize together in the context of long RNAs. For example, the sequences we identified as influential when using the β -globin backbone may synergize in specific ways with the known elements in β -globin that inhibit export²⁸. It is also possible that imperfections in the nuclear/cytoplasmic fractionation procedure, where some residual cytoplasmic RNA occasionally remains associated with the nuclei, limit the ability to detect large effect sizes. Notably, a series of recent MPRNAs applied to neurite/soma fractionations, some using the same experimental design as we do here, occasionally found much larger effect sizes, even though in many cases no short elements could be found to recapitulate the activities of full-length 3'UTRs⁶⁷⁻⁶⁹.”

In addition, as described below, following also the request of Reviewer 2, we now added imaging results for three of the tiles bound and affected by IGF2BP1.

2. The authors make an interesting discovery that the circPVT1 exhibits similar localization patterns to “scrambled” SCR circPVT1. They so strongly believe in this correlation that they cite this as a reason to only test non-scrambled circPVT1 under the results section “Context-specific effects of sequence

elements on RNA stability.” This correlation seems to suggest that the localization mechanism for circular RNAs may be heavily driven by the circular structure, and less so by the specific sequence motifs. However, the authors then proceed to claim that circular RNA localization is heavily driven by sequence, for example when discussing SRSF1. I would like to see the authors explore the role of sequence versus structure more, and more explicitly acknowledge the tension or balance between these two ideas.

The relative contribution of sequence and form is indeed the central point central of our study. To address it more comprehensively, we have now performed an MPRNA using a linear form of circPVT1. As we show in the new Supplementary Figure 3c, the pattern of sequence↔localization patterns in this backbone is more similar to the linear form rather than to the circular ones. This suggests that the form of the RNA (linear/circular) has a greater contribution to the eventual localization of the RNA over specific sequence elements, which fits well with our other results. We now explicitly acknowledge this in the Discussion section (page 29):

“We note that the high correlation between circPVT1 and the SCRCircPVT1 backbones, and the lack of similarity to the linear version of circPVT1, all suggest that the backbone sequence has an overall limited effect on the activity of sequence elements that influence localization and/or stability.”

Supplementary Figure 3 - c Correlation plots for the localization of all tiles in each one of the contexts; Spliced, Unspliced, circPVT1, SCR-circPVT1 and Linear-circPVT1. Color-coded values indicate the pairwise Spearman’s correlations between Nuc/Cyto ratios in each sample.

3. In fact, according to S5A (now Supplementary Figure 6a), SRSF1 drives *greater* nuclear enrichment in the scrambled circRNAs. Ostensibly, scrambled circRNAs should carry few motifs, if any. How can this result be reconciled?

We note that every backbone we use has some baseline localization. On top of this baseline localization, the sequence of the backbone can also influence how individual library sequences act within it (e.g., by forming secondary structures with the library sequences). Please note that the scrambled circRNAs we use are not completely scrambled – the backbone sequence is scrambled while maintaining circularization, but the library sequences are naturally not scrambled. Supplementary Figure 6a shows that library sequences containing SRSF1 binding sites elicit a more nuclear enrichment in both “unscrambled” and “scrambled” backbones, suggesting that in the circular form, regardless of the rest of the sequence of the backbone, SRSF1 increases nuclear retention. We now made this point clearer in the text (page 15):

“Tiles bearing the SRSF1 binding motif were enriched in the nuclear fraction especially when expressed in either of the circular backbone sequences (Fig. 3b and Supplementary Fig. 6a), and became more cytoplasmic upon SRSF1 KD in the circular form, and to a lesser extent in the linear spliced form, although these sequences were generally less nuclear in this context (Fig. 3c-d)”

4. For the results surrounding RBPs, the authors specifically discuss SRSF1, SAFB, IGF2BP1, and IBF2BP2. Firstly, IGF2BP2 is not shown in S5A. Furthermore, how were these selected? The text states that they were chosen based on enrichment – I would like to see more specificity around this statement. Are these the genes with the largest fold change difference? Most significant? Largest difference _and_ are recognizable genes? Without such details, these results feel somewhat cherry picked.

In the course of the analysis of the MPRA results, we considered several parameters for each of the factors profiled by ENCODE, including the enrichment of eCLIP clusters of the protein in tiles that are differentially localized in different contexts, as well as AME motif enrichment analysis, and the availability of reagents (e.g., antibodies) for studying these genes further. Through this integration, which was manual rather than automated, we selected four factors for knockdown experiments, and then continued to follow-up based on the results from the knockdown. We did not attempt to knock down other factors in the course of this study. IGF2BP2 was chosen because of its similarity in binding motif to IGF2BP1. Naturally, it is very likely that other factors also contribute to the differential localization patterns that we report, and we hope that they will be further characterized in follow-up studies. We now mention this in the Discussion section:

“We link the activities of three RBPs, SRSF1, SAFB, and IGF2BP1 to the fates of different subsets of RNA molecules. We note that since we followed up on just four factors in this study, and ENCODE eCLIP analysis suggests many additional factors with enrichment patterns at least as strong as the factors we studied (Supplementary Fig. 8A), it is likely that many additional factors sculpt the context-specific effects on localization.”

Minor points:

- For determining significant enrichment, the authors use a cutoff of 0.3 log fold change; please provide justification for this value.

We use 0.3 as it roughly corresponds to a 25% effect size, which is now also mentioned in the text.

- The authors make a point to compare G/C content of transcripts localizing to different regions. However, it is unclear how the p-values attached to these comparisons are generated. Is this a t-test? Mann-Whitney test? The Wilcoxon rank-sum test is used in other scenarios, which leads me to believe this may be test used here as well, but this is not clearly discussed.

We use Wilcoxon rank-sum test throughout the analyses, and now indicate this in the text.

o Similarly, for the ΔG calculations for Figure 2D (now Supplementary Figure 2c). Additionally, if the authors wish to cite the “spliced” context for the structured-ness, please provide a p-value for that as well.

We added this information: $P=2.4 \times 10^{-10}$ for cytoplasmic tiles vs. all other tiles in the spliced context

- How is ΔG calculated? There may be “standard” method I’m not aware of, but I couldn’t seem to find a detailed description of this method.

We now describe this in the methods section.

- Axes for Figure S4D (now Supplementary Figure 5d) are somewhat confusing – I would intuitively expect that GC content would be on the x-axis, as half life is ostensibly a function of GC content, not the other way around.
- Similar for Figure 4F (now 5f)

We changed Figure S4D (now Supplementary Figure 5d) as requested. For Figure 4F (now 5F) we prefer to keep the figure in the current orientation, as throughout the manuscript, we show changes in the localization typically on the y-axis.

- It appears that higher GC content actually leads to more stability in unspliced transcripts in Figure S4D (now Supplementary Figure 5d); I think the authors should mention this, as this is a more similar match to prior results from Litterman et al.

Our previous wording of this part was not correct in that Litterman et al. found that G/C-rich 3’UTRs decrease RNA stability, which matches our results for spliced hosts, but not for the unspliced ones. We have now changed the sentence accordingly (page 13):

“Consistently with a previous study, using a different reporter system³⁴, we found that higher G/C content was associated with reduced RNA stability, mostly in the spliced linear context, and to a lesser extent in the circular context (Supplementary Fig. 5d).”

- Across several figure panels, the authors vacillate between showing p-values, or using asterisks to indicate significance. Recommend showing p-values for consistency and transparency.

We changed asterisks to p-values throughout the figures.

- Figures 4G, H, and I are confusing. For example, for 4G, it is described as showing change in localization of genes given different numbers of eCLIP clusters. However, the plot itself does not seem to really show the “baseline” unless that baseline is 0 (unclear). Additionally, the statistical testing is unclear to me here. The asterisks seem to indicate significant difference between a group and the union of all other groups that are either lower or higher. This seems like a strange way to do statistical testing. If the takeaway here is that increasing eCLIP clusters increases effect of knockdown, wouldn’t it be more direct to use something akin to a Spearman’s correlation? Or, simply bisect the data by number of eCLIP clusters and compare high versus low? Even if the authors insist on this type of comparison, it seems prudent here to do some sort of multiple hypothesis correction here (which is not mentioned), seeing as they are essentially comparing “gain” and “losses” across 6 different comparisons.

We agree with the reviewer that the previous representation was confusing, and as suggested we revised the analysis to focus on a bipartite comparison between the “CLIP low” and “CLIP high” sets of genes. All the conclusions remain the same. We now moved all these plots to Supplementary Figure 8, as we prefer to focus on the FISH validation results in the main figure.

Supplementary Figure 8 - IGF2BP1 regulates nuclear export of linear spliced RNAs. d Change in localization of transcripts with low (<3) or high (≥10) number of IGF2BP1 eCLIP clusters in K562 cells

following KD of IGF2BP1 in MCF-7 cells. Number of genes in each group is specified below. Only genes expressed in both MCF-7 and K562 cells were considered. **e** Change in expression of genes with low or high number of IGF2BP1 eCLIP clusters (defined as in d) following KD of IGF2BP1. Number of genes in each group is specified. **f** Change in localization of single-exon transcripts (left) and multi-exon transcripts (right) with low or high number of IGF2BP1 eCLIP clusters following KD of IGF2BP1. Number of genes in each group is specified. **g-h** Change in expression of genes with low or high number of IGF2BP1 (**g**) or IGF2BP2 (**h**) eCLIP clusters following KD of IGF2BP1 (**g**) or IGF2BP2 (**h**) in the cytoplasmic and nuclear fractions. Number of genes in each group is specified below. **i** Change in localization of transcripts with low or high number IGF2BP2 eCLIP clusters following KD of IGF2BP2. Number of genes in each group is specified.

Reviewer 2

In this manuscript, Ron and Ulitsky use MPRNA to conduct a broad survey of cis-elements that promote or inhibit the nuclear export of unspliced, spliced and circular reporter RNAs. They find that elements with high GC content promote export of unspliced and spliced RNAs, similar to their previous findings and from those from the Kudla group. Interestingly, this does not appear to affect circular RNAs. Based on these results, they then go on to test whether trans-elements (RNA binding proteins), whose motifs are enriched in nuclear or cytoplasmic RNA pools, affect nuclear export or nuclear retention in MPRNA experiments. From this they find that distinct types of RNA (i.e. unspliced, spliced and circular RNAs) are differentially responsive to different trans-factors. In particular SRSF1 appears to promote the nuclear retention of spliced and circular RNAs, SAFB promotes the nuclear retention of unspliced RNAs, and IGF2BP1 promotes the export of spliced and unspliced RNAs. They then examine how IGF2BP1 and 2 affect the nuclear/cytoplasmic distribution of endogenous mRNAs.

The paper is very interesting and of broad interest to the mRNA community. I am generally in favor of publication as long as some of the following two major concerns are addressed.

1) Much of the paper is based on RNAseq analysis. The results should be validated by imaging data – this should not be too hard as examples of reporters with elements that promote nuclear retention or export can be individually expressed and visualized by fluorescent in situ hybridization (FISH). The authors should provide examples of cis-elements that promote retention, and export. The authors should also validate that the depletion of trans-factors (SRSF1, SAFB and IGF2BP1) alters these distributions. Finally, authors should validate that depletion of IGF2BP1 affects the nuclear/cytoplasmic distribution of particular endogenous poly(A)-mRNAs (as predicted by their analysis) by single molecule FISH.

We agree with the reviewer that adding an imaging component is important. We chose to focus on imaging reporters with specific sequence tiles from our libraries, as those are better controlled experiments, where we can also examine the effects of the knockdown on the “empty” reporters. As visualizing circular RNAs by FISH is a major challenge, we further focused on the factors affecting predominantly the linear forms. We established stable lines expressing the beta-globin RNAs in spliced and unspliced forms with different inserts from the MPRNA libraries.

For the unspliced form, we found that the expression levels in such format are too low to enable a robust quantitative analysis (we note that this setting is different from the plasmid transfection which we use in our MPRAs, which allow both higher expression levels of the reporters and considering millions of cells at one). Therefore, we present smFISH analysis of the localization of the spliced beta-globin form, containing three tiles from our library with IGF2BP1 motifs and/or CLIP peaks. An additional tile resulted in a strong sensitivity to IGF2BP1 for its expression and did not give reliable results for localization. In all three tiles presented (NEAT1#17, NEAT1#81, and Hoxa11-AS #2), we found that siRNA-mediated knockdown of IGF2BP1 resulted in a more nuclear localization of the RNA, consistent with the results of our MPRNAs.

The text describing these results (page 21):

“In order to visualize the effect of IGF2BP1 on RNA export, we established lines of MCF-7 cells stably expressing a Doxycycline-inducible spliced β -globin mRNA that is fused to one of four different tiles (Mlxipl #73, NEAT1 #17, NEAT1 #81, and Hoxa11-AS#2.) We also attempted to establish similar lines for the unspliced β -globin, but its expression levels in induced cells were too low to enable robust quantification. The tiles were selected based on the change in their localization specifically in the spliced and not in the unspliced context (**Supplementary Fig. 9a**), and on the presence of IGF2BP1 binding motifs and/or CLIP clusters (**Supplementary Fig. 9b**). We then used single-molecule fluorescence in situ hybridization (smFISH) ⁵² to visualize the localization of β -globin mRNA with each of the tiles or with an ‘empty’ 3’ UTR in cells treated with control or IGF2BP1-targeting siRNAs. For the Mlxipl #73 tile, knockdown of IGF2BP1 resulted in a strong reduction in expression levels in both the nucleus and cytoplasm, which prohibited robust estimation of Nuc/Cyto ratios. For the two tiles derived from NEAT1, we found that knockdown of IGF2BP1 resulted in a significantly more nuclear localization of β -globin compared to that of β -globin with an empty 3’UTR (**Fig. 5g,h** and **Supplementary Fig. S9c,d**). For Hoxa11-AS #2, adding the tile significantly enhanced cytoplasmic localization in control cells, but this enhancement was attenuated upon IGF2BP1 KD (**Fig. 5g** **Supplementary Fig. S9c,e**). These results suggest that our observations on the positive effects of IGF2BP1 on the export of spliced RNAs extend to another expression system (inducible expression from a reporter integrated by viral infection) and to an imaging-based localization quantification method.”

Figure 5 - g Quantification of Nuc/Cyto ratios of β -globin signal as measured by smFISH of the inserted tiles relative to the WT β -globin sequence. **h** Representative smFISH images of β -globin-NEAT1#17 (red) and DAPI (blue) in control and IGF2BP1-depleted cells.

Supplementary Figure 9 - c-e Representative smFISH images of β -globin-WT (**c**), β -globin-Neat1 #81 (**d**) and β -globin-Hox11-AS #2 (**e**) and DAPI, in control and IGF2BP1 depleted cells.

2) The effect of IGF2BP1/2-depletion on endogenous mRNAs does not seem very significant. How does this compare to depletion of UAP56 or NXF1? Moreover, information on the experiment is inadequate. Was the RNAseq data from multiple independent biological replicates? Are data from replicates consistent? For details, the text refers the reader to Table S4 (I presume that this is "S4 Supplementary Dataset") but the data presentation in that file is opaque. The enhanced effect on the cytoplasmic/nuclear ratio of mRNAs with many eCLIP clusters (Figure 4G) could simply be an effect on long mRNAs rather than short mRNAs.

We agree with the reviewer that the effects of IGF2BP1 on localization of endogenous RNAs are modest, and now indicate this in the text (page 21):

"Notably, changes were typically modest – transcripts from 160 genes became >2-fold more nuclear upon IGF2BP1 knockdown, a number much lower compared to the effects of knockdown of general export factors in MCF-7 cells (Zuckerman et al. 2020)."

We used three biological replicates, as now mentioned in the text, where we also note the correlations between replicates ($R=0.61-0.79$ between Cyto/Nuc ratios in independent replicates). Following the suggestion of Reviewer #1 we revised the analysis of these data and now have a simplified presentation in Figure S8 which just compares genes with a low number of CLIP clusters with those with many clusters. We note that for the most significant trend that we observe - the increase in nuclear accumulation of IGF2BP1 targets, there is no association with transcript length ($R=0.01$ between the change in expression in the nucleus upon siIGF2BP1 and transcript length).

Minor concerns:

- Intro: The authors should include details about circRNA conservation and estimates of function (PMID: 34320353). The idea that circRNAs may simply be byproducts that accumulate as they do not cause too much harm, but could eventually be disruptive is in line with the association of circRNAs with aging and neurodegeneration (PMIDs: 29753875, 34016449).

We added these ideas and the suggested references to the Introduction on page 2:

"The function of the vast majority of circRNAs remains unknown¹⁰, and it is unclear how many circRNAs are functional^{11,12}, but it is likely that those that do carry out influential functions require specific subcellular distribution patterns. Furthermore, it is possible that aberrant circRNA accumulation, which may occur during aging or neurodegeneration^{13,14}, can have localization-dependent detrimental effects."

- Figure S1E: Specify in the figure legend what RT “+” and “-” mean (reverse transcriptase followed by PCR, vs DNA prep followed by PCR?)

We added the information in the legend.

- Figure S1 and S7 (now S3a): The size of the nucleotide standards should be labelled.

Fixed.

- Figure 2D: Although there is a correlation between “folding” (i.e., delta-G) and export, this is what you would expect given the GC content results. Is the amount of folding in export-promoting sequences greater than what one predicts given the correlation with GC content?

The difference in structure is indeed expected given the higher G/C content of the tiles that promote export. We now have examined this explicitly by randomly shuffling the tiles and repeating the comparison of the folding potential. We found that the shuffled sequences showed folding potential largely similar to that of the actual library sequences. We therefore changed the text and the figures accordingly, and moved these results to the supplement (page 7):

*“Additionally, tiles driving cytoplasmic enrichment in the unspliced, and to a lesser extent in the linear spliced forms were predicted to be more structured ($P=7.9\times 10^{-38}$ for cytoplasmic tiles vs. all other tiles in the Unspliced context, $P=2.4\times 10^{-10}$ for cytoplasmic tiles vs. all other tiles in the spliced context, **Supplementary Fig. 2c**), though it is difficult to uncouple the differential G/C content from the potential to form more stable structures, as shuffled sequences of the tiles driving cytoplasmic enrichment were also predicted to be more structured than shuffled sequences of other tiles (**Supplementary Fig. 2c**).*

Supplementary Figure 2 - c ΔG distributions of the tiles and of a dinucleotide-preserving shuffled version of all tiles, enriched in the cytoplasmic fraction (solid line), nuclear fraction (dashed line) or all other tiles (dotted line) in each context. Vertical gray line indicates the median of all tiles in the sample.

- Page 8: “information driving the strong nuclear enrichment of this lncRNA ... has adapted specifically to be effective within human cells” this is speculative – no data presented suggest that the nuclear-localization activity of MALAT1 fragments is a particular adaptation which has occurred in humans.

We agree with the reviewer that this is a speculation, and have changed the sentence accordingly:

“...is broadly distributed throughout the RNA, and has perhaps adapted specifically to be effective within human cells”

- Figure S3 (now Supplementary Figure 4): Fragments from the 3' end of MALAT1 appear to strongly inhibit export of the circRNAs – are these part of the mascRNA region? Or part of the triple helix at the junction between MALAT1 and mascRNA?

We added the position of the mascRNA to Supplementary Figure 4d (the triple helix appears immediately upstream of it). The tiles that inhibit export do not overlap this region, but are rather found downstream of it.

- Litterman 2019a and Litterman 2019b are the same publication.
Fixed.

- Page 12: “In contrast to a previous study, using a different reporter system (Litterman et al., 2019b), we found that higher G/C content was associated with reduced RNA stability, mostly in the spliced linear context.” The Litterman study reports that GC content in the 3'UTR correlates with a reduction in mRNA stability (the opposite of what was stated by the authors).

We corrected the sentence accordingly:

“Consistently with a previous study, using a different reporter system²⁹, we found that higher G/C content was associated with reduced RNA stability, mostly in the spliced linear context, and to a lesser extent in the circular context (Supplementary Fig. 5d).”

- siRNA sequences should be provided.

We now report those in Supplementary Table 2.

- The effects of SRSF1 on nuclear retention are surprising, although the data seems to indicate that the effect seems modest at best (Figure S5D). How does this compare to changes in export when nuclear export factors are depleted (e.g. in Zuckerman et al., 2020)?

In this case, it is impossible to directly compare the effect sizes, as in Zuckerman et al., 2020 we examined the localization of endogenous RNAs, where we could also look at absolute localization levels (by normalizing to total RNA). In this case, we are looking at only the library sequences, for which normalization to obtain absolute changes is challenging, and so we focus on the differences between *which* tiles affect localization, rather than on absolute changes in localization.

- There are some inconsistencies – in Figure 3F (now 4a) the correlation between SAFB binding motifs and nuclear retention seems similar in spliced and unspliced, while in Figure 3G (now 4b) the white box plots in these two data sets look different. Could the lack in change in localization of spliced mRNAs for SAFB knockouts be due to inconsistencies in the control (NT) data set (comparing 3F and 3G)?
- Similar to the previous point, the correlation between IGF2BP1 binding motifs and nuclear localization of circular RNAs for control-depleted cells looks different between Figure 4A (now 5a) and 4B (now 5b) (no correlation in 4A, nuclear retention in 4B).

There is indeed some variability between the libraries prepared using untreated cells and those prepared from cells transfected by siNT, but we note that the trends in them are largely consistent, in particular the differences we observe upon factor knockdown are consistent. Specifically, for SAFB binding motifs (Figure 4a vs. 4b) there is a trend of increased nuclear localization with a higher number of SAFB binding motifs in both untreated and siNT cells. For IGF2BP1 binding motifs (Figure 5a vs. 5b), there is increased export with increasing number of binding motifs in the linear context, and increased nuclear localization of circular RNAs in both untreated and siNT cells.

- The axes should be as clear as possible to help the reader interpret the figures (i.e., “change in expression” and “change in localization” should be more specific – see figures 4C, 4F, and S5D).

We increased the size of the font, and changed the color of x axis and y axis labels in relevant plots.

- Page 20 “... more than 1 SRSF1 eCLIP clusters ...” this should be IGF2BP1.

Fixed.

- Page 20/21 “Change in localization of genes ...” this should be “Change in localization of RNAs from genes ...” – similarly “Change in localization of single-exon genes” should be “Change in localization of RNAs from single-exon genes”.

Fixed by referring to “transcripts” instead of “genes”.

- Discussion: beta-globin has cis-elements that inhibit export (PMID: 26362019) could inserted elements disrupt or enhance these? Likewise Pvt1 appears to contain many elements that promote export (Figure S3A) - are these enhanced or disrupted by some of the inserts? The authors should discuss these points.

We now mention the potential synergy with the known sequence elements in β -globin in the Discussion section:

“One limitation of our study is that the sizes of the effects on Nuc/Cyto ratios elicited by the individual tiles are typically quite modest and rarely exceed 2-fold. Such effects are comparable to those observed in other studies using a similar system^{22,23,62}. These effects could stem from the limited potency of individual short sequences that typically act in the context of much longer RNAs, and it is unclear how individual short sequences synergize together in the context of long RNAs. For example, the sequences we identified as influential when using the β -globin backbone may synergize in specific ways with the known elements in β -globin that inhibit export²⁸.”

The Pvt1 in Figure S3A (now Supplementary Fig. 4a) refers to the linear form of Pvt1, which contains many sequences that are not within circPVT1 (which encompasses just one PVT1 exon).

- Discussion: Many documented nuclear retention elements appear to be long (e.g. PMIDs: 22355166, 26362019) - are these composed of many smaller elements that act in an additive manner, or do these form larger active units that lose nuclear retention activity when subdivided? The authors should discuss this.

We now mention this question of synergy between short elements and include the suggested references in two places in the Discussion section, one mentioned above, and here:

“One possible reason for this difference is the use of tiles of different lengths in different studies, as some known localization elements are rather long^{28,31}, and so subsequences of different sizes may vary in their ability to recapitulate the activity of the full-length sequence.”

Reviewer #3

In this work the authors used massively parallel RNA reporter assays to study how sequences derived from lncRNAs or circRNAs affect the subcellular localization and stability of linear and circular RNA transcripts. They also performed in-depth analyses of selected RBPs (SRSF1, SAFB, IGF2BP1/2) to study their effects on transcript localization and stability. Overall, this work generated a large amount of data, with some interesting and potentially important observations. However, the present manuscript has some limitations and minor issues that need to be addressed in a revision.

1. Conceptually, the most interesting observation from this work is that the same sequence elements could have distinct impacts on transcript localization and stability, depending on whether they are in linear or circular RNA transcripts. However, there is not much mechanistic investigation or discussion about this observation. At minimum, the authors should expand the Discussion section to offer a more detailed possible mechanistic interpretation of this finding. Also, to what extent can this difference be attributed to sequence difference between the backbone vectors used for the linear vs circular RNA reporter?

For the first point, we have expanded the discussion section to address the potential mechanisms of action (page 25):

“The mechanism of this importance of context remains unclear and will likely be elucidated in future studies. We recently showed that unspliced RNAs exhibit drastically different dependencies on canonical mRNA export factors for their nuclear exit²⁵, as splicing enables a ‘bypass’ of the requirement for folded G/C-rich sequences enabling export of unspliced RNAs, likely through direct or indirect recruitment of NXF1. This may explain the differences we observe in contribution of individual sequence elements to export of spliced or unspliced RNAs, as sequences that facilitate efficient NXF1 recruitment may have a larger influence when placed in unspliced RNAs and only a marginal contribution in spliced ones. For the differences between linear and circular form, one possibility is that the export of linear RNAs is heavily influenced by proteins binding to the 5' cap and to the poly(A) tail, which are absent from circular RNAs, which may have to recruit such factors indirectly. Another aspect to consider is that circular RNAs can adopt more compact folds compared to linear forms of the same sequence, as was recently demonstrated⁵⁴, which can influence the ability of RBPs to access and bind specific sequences, and thus influence RNA export.”

To address the second point, following related suggestions by Reviewer 1, we have not performed an MPRNA using a linear form of the circPVT1 sequence, which we found to behave more similarly to the linear beta-globin forms than to circPVT1, suggesting a dominant effect of the form of the RNA over its sequence. These new data are presented in Supplementary Figure 3c:

Supplementary Figure 3 - c, Correlation plots for the localization of all tiles in each one of the contexts; Spliced, Unspliced, circPVT1, SCR-circPVT1 and Linear-circPVT1. Color-coded values indicate the pairwise Spearman's correlations between Nuc/Cyto ratios in each sample.

2. The present work is largely based on massively parallel RNA reporter assays. It would be helpful if the authors can perform more in-depth studies to see if and how patterns learned from the reporter assays can be translated into endogenous genes. There are some efforts to do this (e.g. sequencing and analysis of polyA+ RNAs upon IGF2BP1/2 knock-down) – but I think this kind of analysis should be expanded. For example, while the authors performed qPCR analysis of 10 selected circRNAs in nuclear and cytoplasmic fractions following SRSF1 knockdown (Fig. 3E), they should carry out a more comprehensive analysis by performing circRNA sequencing of these RNA samples.

As suggested by the reviewer, we have now performed sequencing (using the RPAD protocol that enriches for circular RNAs) on nuclear and cytoplasmic fractions of MCF7 cells after knocking down SRSF1. Consistent with our MPRA results, this unbiased analysis shows that SRSF1 binding is associated with nuclear enrichment of endogenous circRNAs, and that SRSF1 knockdown results in a cytoplasmic shift in localization of these RNAs. These results are presented in Figure 3f and Supplementary Figure 7c-e:

Figure 3 - f Change in Nuc/Cyto ratio for circRNAs with the indicated number of SRSF1 binding motifs. P-values are for comparing with circRNAs with no motifs and were computed using a two-sided Wilcoxon rank-sum test. Number of circRNAs in each group is indicated in parentheses.

Supplementary Figure 7 - c Baseline Nuc/Cyto ratios for circRNAs annotated in circBase with the indicated number of SRSF1 binding motifs. P-values computed using Wilcoxon rank-sum test, and the number of circRNAs in each group is indicated in parentheses. **d,e** Mean number of SRSF1 binding motifs (**d**) or eCLIP clusters (**e**) in the sequences of circRNAs with the indicated change in localization following KD of SRSF1. P-values computed using two sided t-test.

3. The authors seemed to have used ENCODE RNA-seq data of subcellular fractions in various analyses (one example being Fig. 3I), but there is no description in the Methods section about where these data came from or how they were analyzed.

We added this information to the Methods section:

ENCODE RNA-seq data analysis

RNA-seq libraries from nuclear and cytoplasmic fractions of HepG2 cells from the ENCODE project were quantified using human RefSeq annotations using RSEM (Li and Dewey 2011) and intersected with ENCODE eCLIP clusters as in (Lubelsky and Ulitsky 2018).

4. Fig. S6, panel B legend. 'SRSF1' seems to be a typo for 'IGF2BP2'.

Fixed.

5. Fig. S7 in the Discussion section describes primary data, thus should be moved to the Results section.

We now moved these results to be presented in Supplementary Figure 3, and describe them in the middle of the Results section (page 10).

6. Methods section – can the authors elaborate how RNA stability was calculated from the actinomycin D treatment data?

We added this information to the Methods section.

REVIEWERS' COMMENTS

Reviewer #1 (Remarks to the Author):

The authors have largely addressed my concerns. I support publication.

Reviewer #2 (Remarks to the Author):

The authors have addressed most of my concerns. I just have a few comments.

- The FISH images (Figure 5H and Supplementary Figure 9C-E) are very hard to see and evaluate given that they are pseudo-coloured red on a black background. I would encourage the authors to display the FISH images in grayscale, while keeping the FISH/DNA overlay in red/blue.
- The microscopic images (Fig 5H & S9C-E) are missing scale bars.
- Figure 5 is duplicated.
- P13 "Consistently with a..." should be "Consistent with a ..."

Reviewer #3 (Remarks to the Author):

The authors have addressed my comments in this revision.

Reviewer #2 (Remarks to the Author):

The authors have addressed most of my concerns. I just have a few comments.

- The FISH images (Figure 5H and Supplementary Figure 9C-E) are very hard to see and evaluate given that they are pseudo-coloured red on a black background. I would encourage the authors to display the FISH images in grayscale, while keeping the FISH/DNA overlay in red/blue.

Fixed.

- The microscopic images (Fig 5H & S9C-E) are missing scale bars.

Fixed.

- Figure 5 is duplicated.

Removed.

- P13 "Consistently with a..." should be "Consistent with a ..."

Fixed.